# Oradea Metropolitan Area as a Space of Interspecific Relations Triggered by Physical and Potential Tourist Activities

Corina-Florina Tătar, Iulian Dincă *, Ribana Linc, Marius I. Stupariu, Liviu Bucur, Marcu Simion Stașac and Stelian Nistor

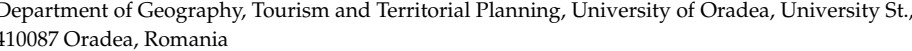

Department of Geography, Tourism and Territorial Planning, University of Oradea, University St., 410087 Oradea, Romania
* Correspondence: idinca@uoradea.ro or iulian_dinca@yahoo.co.uk

**Abstract:** Metropolitan areas provide many opportunities to spend quality outdoor leisure time as well as to discover many cultural attractions. Sprawl occurs in Romania quite rapidly, encouraged by the construction of ring roads around many cities and their expansion into metropolitan areas. The current paper aims to identify metropolitan tourism models based on which tourist flows can be sustainably reoriented within rural Oradea Metropolitan Area (OMA) given their the tourist potential level (i.e., very low, low, average, high). The tourist potential was scaled based on the Methodology for the Analysis of a Territory's Tourist Potential, which stands as a law published in the Official Monitor of the 14th of June 2016. The study indicates that most tourist activity develops in the OMA southern part in Sânmartin commune, thus unsustainably capturing all tourist flows of the rural OMA. Natural and man-made tourist attractions' territorial concentrations were emphasized in the communes from the south and northern OMA, but there are major territorial dysfunctions in terms of technical endowment and tourist infrastructure supply. The three emerged models refer to the medical–recreational and eco–residential wellness network, discovery eco-holiday, and co-visit and marginal community.

**Keywords:** Oradea metropolitan area; tourism potential; assessment; supply; demand; tourism models

## 1. Introduction

Tourism can contribute to sustainable development in the perpetually expanding urban areas by balancing economic, social and environmental goals therefore tourist potential knowledge and scaling is important for planners and policy-makers based on legal frameworks [1]. In the future, it is estimated that 80% of the global population will live in cities and only 20% in rural areas, a trend that had already started in 2008, when the number of people living in cities exceeded the number of those living in rural areas [2]. The growing urban expansion has yielded metropolisation.

Many suburbs have turned into genuine dormitory satellites around big cities for a commuting population. Through metropolisation, the urban and rural have intermingled, and this has created the right premises for the commuter metropolitan inhabitant to be in frequent contact with nature. At the outskirts of urban agglomerations, tourism development has become very dynamic and metropolisation seems to strengthen this phenomenon [3]. The metropolitan area provides many opportunities to spend quality outdoor leisure time as well as many opportunities to discover cultural attractions such as ethnic neighbourhoods, historical attractions, etc., as well as events generated by the cultural attractions such as fairs, festivals, etc. [4]. In addition, metropolitan tourism is more likely to occur for a family due to the metropolitan attractions' accessibility in terms of distance and time, which would reduce the financial strain allocated to travelling. Most metropolitan attractions are reachable within a day, so both commuters and urbanites can take day trips that are efficient in terms of cost and time for visitation, as postulated by the distance-decay concept, in which the closer the attractions, the more prone for visitation [5].

The proliferation of satellite cities, the conversion of seasonal secondary settlements into full-time residencies at the urban fringe, led to metropolisation. Most post-socialist urban Eastern European cities feature large socialist-era housing near the inner-city periphery; nonetheless, the metropolitan development pattern protected agricultural land and offered modest high-density housing for workers, and these cities are now experiencing rapid commercial increase and residential growth at the urban fringe [6]. Sprawl occurs in Romania quite rapidly, encouraged by the construction of ring roads around many cities, as well as investor preferences for greenfield rather than brownfield development. Since 2001, there has been a general expansion of residential areas in all suburban communes, despite demographic decline [7]. The rural metropolitan areas, leisure peripheries [8]; peripheral areas [9], which are designated in connection to tourism activities; peripheral rural space [10]; and rural periphery [11], as they are also terminologically known in specialized literature, offer good potential for the development of tourism.

The tourist potential is the tool by which specific tourist models are elaborated within the current paper. The tourist potential is not unitarily defined among authors, but rather used generically to apply to more tourism research areas. Bakhodirovna (2021) [12] approaches it in terms of cultural and historical values and the quality of services in the fields of ethnic, ecological, tourism, and the development of rural and gastronomic tourism, whereas Pralong (2005) [13] looks at it from a scenic, scientific, cultural, and economic value for the tourist potential assessment of geomorphological sites. The geosite potential was also carried out by Pál and Albert (2018) [14]. Tourist potential is largely used as terminology in Romanian tourism geography books [15], and, according to some authors, it refers to the natural and cultural man-made resources [16], whereas other Romanian authors [17], besides the natural and cultural resources, also include the tourist facilities or infrastructure. Finally, the World Tourism Organization (UNWTO) defines it as the sum of the natural, anthropogenic, and material resources and conditions necessary for destination management [18]. Furthermore, tourism potential refers to the natural, cultural, historical, and socio-economic background for the organization of tourist activity in the particular area [19]. Tourism potential is approached not only from a resource based perspective but also in terms of facilities and infrastructure to make attractions visitor-ready. Destinations can be at any scale, from a whole country to a village. The attractiveness of a destination can be approached from the demand or supply side, the latter referring to the number and quality of available attractions at destination [20]. Buhalis (2001) [21] refers to the supply side as a factor of competitiveness. It is important to make an assessment of resources for helping local governments make decisions on allocating resources for sustainable tourism development with a view to tourism planning [22], in order to determine the value of attractions and hierarchize them [23] or as a necessary step in order to know the potential of relevant resources before marketing the destination [24]. Thus, the tourism potential assessment is important with a view to planning, publicity, investment, and management. Tourism potential can be divided into two categories, natural and cultural, and it has been analyzed across time by more authors such as Du Cros (2001) [25]; McKercher and Ho (2006) [23]; Ptacek et al. (2015) [26]; Yan et al. (2017) [20]; Mamun and Mitra (2012) [27]; Bucurescu (2013) [28]; Collins-Kreiner and Wall (2007) [29]; Emphandu and Ruschano (2007) [30]; Kresic and Prebezac (2011) [31]; Sanchez Rivero et al. (2016) [32]; Shohan et al. (2012) [19]; and Hoang et al. (2018) [33].

The prevalently used model for assessing tourist resources lies in Du Cros's (2001) model, mainly applied for assessing the cultural heritage [34], such as performed by Li and Lo (2004) [35]. Another model refers to the item response theory, where Sanchez Rivero et al. (2016) [32] used this model to weight the qualitative and hierarchical evaluation attributes to evaluate and rank resources. McKercher and Ho model (2006) [23] relates to the cultural, physical, product, and experiential values of assets. Other models refer to stakeholders' assessment [36]; GIS tool as a means to evaluate tourism resources [37]; and the weighted sum model [38], a mathematical model for assessing the tourist potential as developed by Yan et al. (2017) [20], meant to audit the heritage of large quantities which consist of two

indicators, such as resources value and development state, so that an assessed site can be scaled into low, medium, and high potential.

The model applied within the current paper for the territory of the Oradea metropolitan area is part of a National Territorial Planning of Romania Strategy as well as of the Romanian legislation published in the Official Monitor of the 14th of June 2016. The tourist potential is assessed in terms of the natural, man-made, tourist, and technical facilities perspectives and is meant to hierarchize certain regions over others and scale them into high, average, and low potential, thus strengthening the idea of a territory's competitiveness [1]. This model can be applied to any scale territory and it was used by Iațu and Bulai (2011) [39] for assessing the tourist attractiveness of the historical province of Moldavia, Tătar (2009) [40] and for rating the attractiveness index of the territory found alongside the catchment area of Crisul Repede River of western Romania. Other models used in the Romanian literature for assessing the potential having as a base ground the aforementioned national methodology are those of Dezsi, Şt. (2008) [41]; Voicu F. (2019) [42]; and Cianga et al. (2002) [43].

## 2. Contextualization

The sprouts of the Oradea metropolitan area (OMA) (the only metropolitan area of Western Romania) were set up along with the ring road of Oradea City for traffic decongestion in 2004. Since then, many metropolitan transportation ways were created and improved in order to feed the growing needs of the metropolitan commuter citizen, and the city has developed as a polycentric metropolis based on an open local economy, multi-firm, and linked by ties that create more or less polarized areas, as well as external ties that have become decisive for its growth. The old industries on the city fringe have been replaced by new sectors such as electronics, business, and services [44]. Certain corridors are thus created, meant to improve the living standards of its residing metropolitan communities [45] and enhance the prosperity of the metropolitan territory. All the necessary principles are implemented to ensure a territorial cohesion by direct investments and high-impact major projects for the area's sustainability.

The study area relates to the rural part of the Oradea metropolitan area, which is an association set up in 2005 and includes the eleven constituent communes of Biharia, Borș, Cetariu, Ineu, Nojorid, Oșorhei, Paleu, Sânmartin, Sântandrei, Girișu de Criș, Toboliu, and the municipality of Oradea city, the latter alone counting 201,547 inhabitants at its core. Each commune is focused on a specific development corridor, such as in Borș commune, where we encounter more industry and agriculture [46], while in Sânmartin, the focus is on geothermal and treatment facilities [47]. Therefore, tourism is more developed in the southern part of the metropolitan area, but on a more detailed tourist supply analysis, the entire rural metropolitan area features propitious premises and resources for tourism. The rural OMA tourist supply holds 254 accommodation units, most concentrated in the southern part of the metropolis, namely in the two resorts of Băile Belix and 1 Mai of the Sânmartin commune with a share of 93.3% [48]. In terms of area, the largest OMA commune is Nojorid and the most populous is Sânmartin (Figure 1).

The rural OMA holds favourable premises and potential for tourism such as thermal waters, vineyards, old churches, and handicrafts centres which can lure urbanites from Oradea as well as international tourists from Hungary, Germany, Israel, The Republic of Moldavia, etc. [49]. Within the Urban Development Integrated Strategy of the Oradea Municipality [50] it is stated that there are about 25,000 daily commuters between Oradea City and the rural metropolitan belt. Most rural metropolitan belt localities are reachable through the public and private transportation within 15–20 min and the highest OMA transit traffic flow is regularly recorded in the communes of Oșorhei, Bors, and Sânmartin communes, accounting for the fact that all the communes are crossed by important roads that connect bigger localities of Romania.

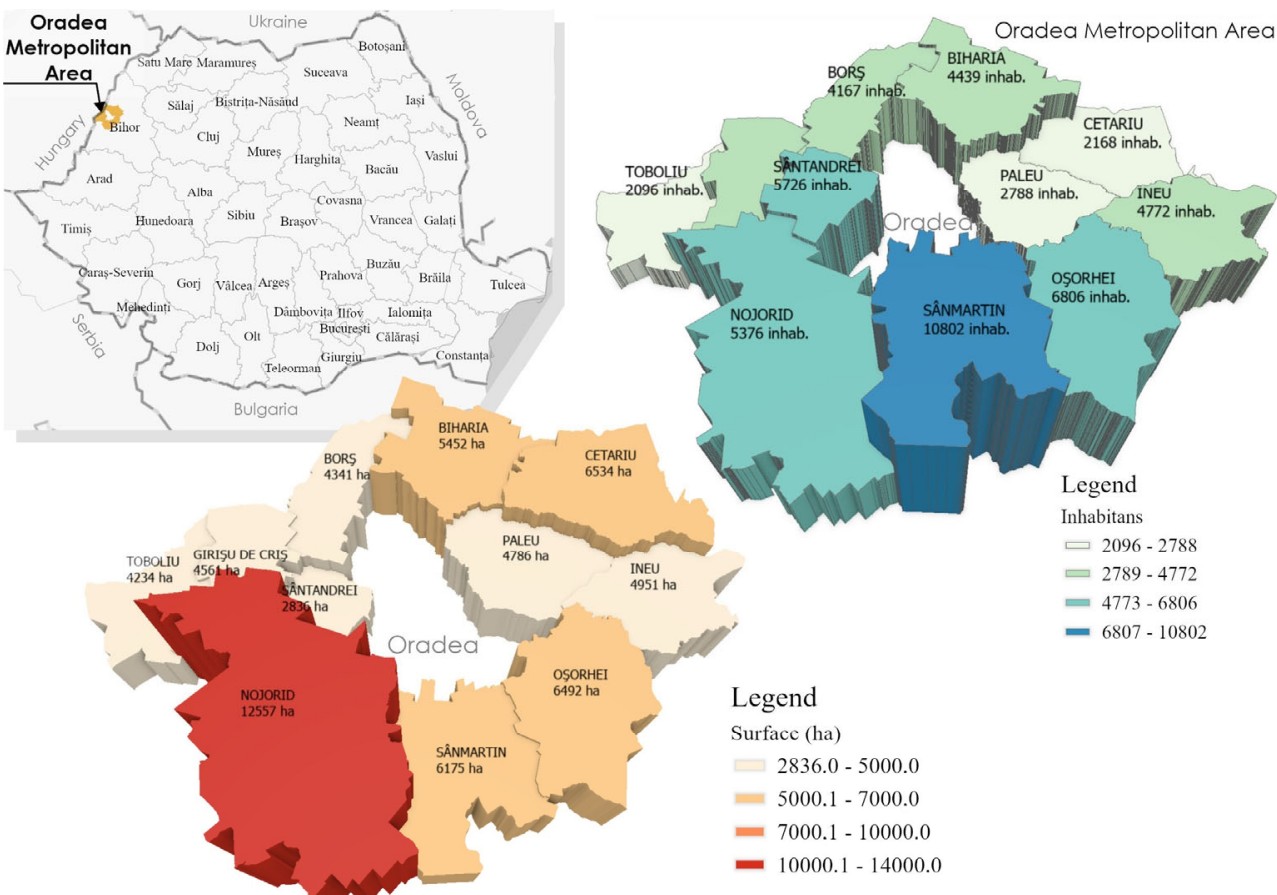

**Figure 1.** Oradea Metropolitan Area constituent communes spread and population and its location within Romania.

Mobility among communes has increased gradually since 2005, when the OMA was set up and many surrounding communes were preferred by former Oradea urbanites as their main residence, but who are still commuting for work in Oradea City. They were chosen for their higher quality of life, outdoor nature, and generally greener habitat. One of the main goals of OMA is to create a functional green infrastructure either at the level of Oradea municipality or at the level of each commune. The next step will be the establishment of an integrated green infrastructure which will fulfil the aim of national and European green policy [50,51].

A way forward to a sustainable approach towards commuting in the OMA are electric vehicles so that the society can use and benefit from an environmentally friendly transport system [52]. In a study conducted by Zientara et al. [53], it was shown that the most educated tourists were more likely to opt for public transport and that walking around is a preferred way of mobility. In the OMA, there are about 25,000 daily commuters who can contribute to a low carbon footprint by the use of public transport and electric vehicles, and further planning of OMA interconnecting concentric and linear bicycle lanes, which is unfortunately yet rather sequential and disrupted.

The means of transport plays an important role in tourism development. Despite the situation of Romania's means of transport, in terms of density, technical background, accessibility, and interconnectivity, is relatively poor compared to the European standards, the situation of OMA, close to the Romanian–Hungarian border, gives a strong point in terms of transport accessibility [54].

Tourism activity in the rural metropolitan area of Oradea is basically concentrated in its southern part, which makes tourist flows very uneven and therefore disproportionalities tend to increase with each passing day. More demand will be placed on the southern part's natural environment, which requires a durable solution to an inevitable challenge to ensure future sustainable tourism metropolitan development [55]. In the local authority's strategy, there is a gap between sustainable development theories and their implementation; the tourist product needs protection, and unsustainable tourist flows and congestion directed towards one commune of the metropolitan belt puts pressure on the topography, nature, and attractions, increases litter and pollution, and puts strain on the local infrastructure, eventually leading to a major reduction of the visitor's enjoyment and appreciation of the site [56].

The current study synthesizes the results of previously published research papers on the topics of nature [57,58], man-made [59] tourist facilities and technical supplies [48], and tourist flow [60] within the rural OMA area, but its original value is that it makes a synthesis of all four categories of nature and man-made attractions and the tourist and technical infrastructure/facility, individually approached in previous studies. The tourist attractiveness ratings for each of these categories is displayed in Table 1. These ratings helped with the elaboration of the current study's tourist potential maps from all these four perspectives into an integrated study. The methodology for obtaining these ratings is featured in the chapter below. The OMA case analysis is necessary because it shows the disparities among communes in terms of tourist potential, thus highlighting a hierarchization of communes at a local level as well as proposing tourist consumption models based on the natural or man-made heritage that each constituent commune possesses. It is a local-based case study which helps create an overall understanding of the OMA's current potential for tourism development, and through the proposed models, a profiled tourist consumption orientation.

**Table 1.** The rural OMA tourist potential assessment.

| Assessed Category/Commune | Biharia | Borș | Cetariu | Girișu de Criș | Ineu | Nojorid | Oșorhei | Paleu | Sânmartin | Sântandrei | Toboliu | OMA Total | Average |
|---|---|---|---|---|---|---|---|---|---|---|---|---|---|
| A1. Natural resources | 3.75 | 4.5 | 6.1 | 4.5 | 5.5 | 5.5 | 5.85 | 5.5 | 5.95 | 4.5 | 4.5 | 56.15 | 5.10 |
| A2. Cure factors | 0.22 | 0.72 | 0.72 | 0.72 | 0.22 | 0.72 | 0.22 | 0.22 | 2.22 | 1.22 | 0.22 | 7.42 | 0.67 |
| A3. Nature tourist reserves | 0 | 3 | 1 | 3 | 0 | 3.3 | 3 | 3 | 3.5 | 1.8 | 2.3 | 23.9 | 2.17 |
| **Natural attractions category subtotal (max. 25 p)** | **3.97** | **8.22** | **7.82** | **8.22** | **5.72** | **9.52** | **9.07** | **8.72** | **11.67** | **7.52** | **7.02** | **87.47** | **7.95** |
| B1. National interest historical monuments | 2 | 0.5 | 4 | 2 | 2 | 6.5 | 2 | 0 | 6 | 4 | 2.5 | 31.5 | 2.86 |
| B2. Museums and public collections | 0 | 0 | 0 | 0 | 0 | 0 | 0 | 0 | 0 | 0 | 0 | 0 | 0.00 |
| B3. Folk craft and tradition | 6 | 6 | 6 | 4 | 1 | 1 | 3 | 6 | 6 | 4 | 2 | 45 | 4.09 |
| B4. Concert and Show Halls | 0 | 0 | 2 | 0 | 0 | 2 | 0 | 0 | 0 | 4 | 0 | 8 | 0.73 |
| B5. Repeated cultural manifestations | 0 | 0 | 0 | 0 | 0 | 0 | 0 | 0 | 0 | 0 | 0 | 0 | 0.00 |
| **Man-made attractions category subtotal (max. 25 p)** | **8** | **6.5** | **12** | **6** | **3** | **9.5** | **5** | **6** | **12** | **12** | **4.5** | **84.5** | **7.68** |
| **Category subtotal of nature and man-made attractions (max. 50 p)** | **11.97** | **14.72** | **19.82** | **14.22** | **8.72** | **19.02** | **14.07** | **14.72** | **23.67** | **19.52** | **11.52** | | |
| C1. Accommodation facilities | 1 | 2 | 1 | 0 | 1 | 1 | 1 | 1 | 7 | 1 | 0 | 16 | 1.45 |
| C2. Cure facilities | 0 | 0 | 0 | 0 | 0 | 0 | 0 | 0 | 5 | 0 | 0 | 5 | 0.45 |
| C3. Conference and exhibition rooms | 0 | 0 | 0 | 0 | 0 | 0 | 0 | 0 | 5 | 0 | 0 | 5 | 0.45 |
| C4. Skiing slopes and cable transportation | 0 | 0 | 0 | 0 | 0 | 0 | 0 | 0 | 0 | 0 | 0 | 0 | 0.00 |
| C5. Other entertainment facilities | 1 | 0 | 0 | 0 | 1 | 0 | 0 | 1 | 1 | 0 | 0 | 4 | 0.36 |
| **Tourist facilities category subtotal (max. 20 p)** | **2** | **2** | **1** | **0** | **2** | **1** | **1** | **2** | **18** | **1** | **0** | **30** | **2.73** |
| D1. Direct access to major transportation | 7.5 | 7.5 | 0 | 0 | 0 | 12.5 | 10 | 0 | 7.5 | 0 | 0 | 45 | 4.09 |
| D2. Utilities infrastructure | 7 | 9 | 2.5 | 0 | 2.5 | 3.5 | 2.5 | 5 | 5 | 6.5 | 0 | 43.5 | 3.95 |
| D3. Communication ways Communication ways | 3 | 3 | 3 | 3 | 3 | 3 | 3 | 3 | 3 | 3 | 3 | 33 | 3.00 |
| **Technical facilities category subtotal (max. 30 p)** | **17.5** | **19.5** | **5.5** | **3** | **5.5** | **19** | **15.5** | **8** | **15.5** | **9.5** | **3** | **121.5** | **11.05** |
| **Tourist and technical facilities category subtotal (max. 50 p)** | **19.5** | **21.5** | **6.5** | **3** | **7.5** | **20** | **16.5** | **10** | **33.5** | **10.5** | **3** | | |
| **All Categories Total Score (max. 100 p)** | **31.47** | **36.22** | **26.32** | **17.22** | **16.22** | **39.02** | **30.57** | **24.72** | **57.17** | **30.02** | **14.52** | **323.47** | **29.406** |

Source: own elaboration based on Stașac et al., (2020), Tătar et al., (2018), Tătar et al. (2021) [48,57,59].

### 3. Materials and Methods

The field research was conducted based on the transferable methodology to any small or large-scale area, which addresses case studies in particular, entitled Methodology of the 25 April 2016 for the Analysis of a Territory's Tourist Potential [1].

The aim of the current paper is to identify metropolitan tourism models which can be used in order to reduce human pressure driven by tourist flows, and thus redistribute the latter based on the tourist potential level (i.e., high, average, low, or very low) of the OMA commune. In order to accomplish this, we followed some objectives such as the synthesis of the natural, man-made, tourist, and technical facilities tourist potential.

The first work hypothesis is that there is a rural tourist OMA hotspot with a high concentration of nature and man-made attractions. The second hypothesis starts from the premise that all communes provide a tourist supply for a segmented market, but which is unevenly dispersed in the OMA communes. The third hypothesis is that there are tourist concentration poles of the tourist supply, as well as intercommunal dysfunctions in relation to the visitor attractions and technical facilities. The synthesis and assessment of the tourist potential relies on the previously mentioned methodology [1], a legal act conceived by the Romanian Ministry of Regional Development and Public Administration. Thus, the tourist potential is inventoried from the angle of the natural and cultural resources, as well as from the tourist facilities and technical endowment. According to this national methodology, the tourist potential is partitioned and scored into 4 large categories: natural resources with 25 points; cultural resources with 25 points; the specific tourist infrastructure/facilities with 20 points; and the technical infrastructure/facilities with 30 points, totalizing a maximum of 100 points as the highest expression of tourism development. Since it is a quantitative research method, it can be transposed to any spatial scale.

The national methodology applied for this study allowed a scoring and assessment of the identified attractions from the OMA communes. Each category comprises certain relevant items, such as for the natural resources, and the targeted items are related to the relief, geomorphology, vegetation and fauna, hydrography, and landscape. For assessing a natural resource pertaining to a certain relief shape, the score assigned to that resource increases with the spectacularity of the relief shape where the resource is located. If the natural resource is found in the plain area, the score is lower, i.e., 1 point; if it belongs to the hilly area, it gets 2 points, and if it belongs to the mountains, it gets 3 points. The second main category of the man-made attractions assesses attractions such as historical monuments, folk crafts and traditions, and institutions of concerts and cultural events, each subcategorized into further specific items and assigned individual scores so that each subcategory of the man-made resources totals a maximum of 8, 9, or 4 points respectively, with a total of 25 points for the entire man-made resources category. In case such attractions existed, they were given the allocated points, and their absence was scored with 0 points. The specific tourist infrastructure category targets accommodation facilities, treatment facilities, conference spaces, and entertainment facilities which are scored with a maximum 7, 5, 6, and 1 points. For assessing the technical infrastructure category, a maximum of 30 points are given for the proper transport access to the identified attraction, access to utilities, and telecommunication networks access. Attractions were identified based on field research between June 2018 and January 2019, based on which a dataset was elaborated, managed by the ArcGIS Pro 3.0.2 by Esri, to illustrate the assessed rural tourist OMA potential shown in the themed maps.

Table 1 indicates which subcategories are comprised within each of the four main categories and each category's total for every individual OMA metropolitan commune. The study results were interpreted according to these four criteria. These scores were attained according to specific analysed items and published within previous research papers. By this assessment and the outcome of the criteria scores, each commune's tourist valences can be easily perceived by comparison to the nearby one, highlighting its tourist potential.

The secondary data have been gathered from previously published research papers and calculated in percentages so as to have a holistic image of the entire development

potential of the rural OMA territory in the synthesis map and broken down into particular segments for the natural/man-made potential and for the technical/tourist facilities. We have split the 100% score into equal intervals, featured in the themed maps legend. Thus, the OMA communes possessing a very low potential have intervals with values between 0 and 20%; a low potential has values between 20.1 and 40%; an average potential has values between 40.1 and 60%; a high potential has values between 60.1 and 80%; and a very high potential has values between 80.1 and 100%. These values illustrate, by comparison, which commune of the rural OMA is best endowed from the point of view of attractions and facilities for tourism development.

The current paper Is structured on two panels. The first is the concentration of the natural and man-made tourist resources, and the second panel is related to the dysfunctionalities of the technical and tourist facilities.

## 4. Current Research Results and Discussions

From the analysis of all categories that were assessed referring to the natural, cultural, and accommodation supply and the technical endowment, Table 1 below illustrates the potential for tourism development of each individual commune and a tourism potential average was calculated for the rural OMA area.

From the analysed tourist-related items, two main directions of the current paper can be determined, namely the natural/man-made attractions' concentration and the dysfunctionalities of the tourist and technical facilities.

For the former related to natural attractions' supply, it came out that the most endowed is the commune of Sânmartin with 46.7% or 11.67 points in absolute values, and the least endowed is Biharia commune (Table 1, Figure 2). The gap is given by the presence of natural factors (mainly, its thermal waters and therapeutic muds) and the presence of natural areas found in Sânmartin commune, placing it in the top of the category, whereas, at the opposite end, is Biharia, which is deprived of such resources and is bottom-placed. Thus the commune of Sânmartin has an average potential in terms of natural attractions, whereas Biharia has a very low potential. We are not going to insist upon these attraction features and the way in which the attractions scores were allocated, as they are extensively dealt with within the study of Tătar et al. (2018) [57] and briefly summarized at the section Previous tourism research results of the constituent communes of the Oradea Metropolitan Area of the current study. The average of the rural metropolitan area is of 7.95 points which situates it at a low (towards average) score compared to the 25 value points granted to this category.

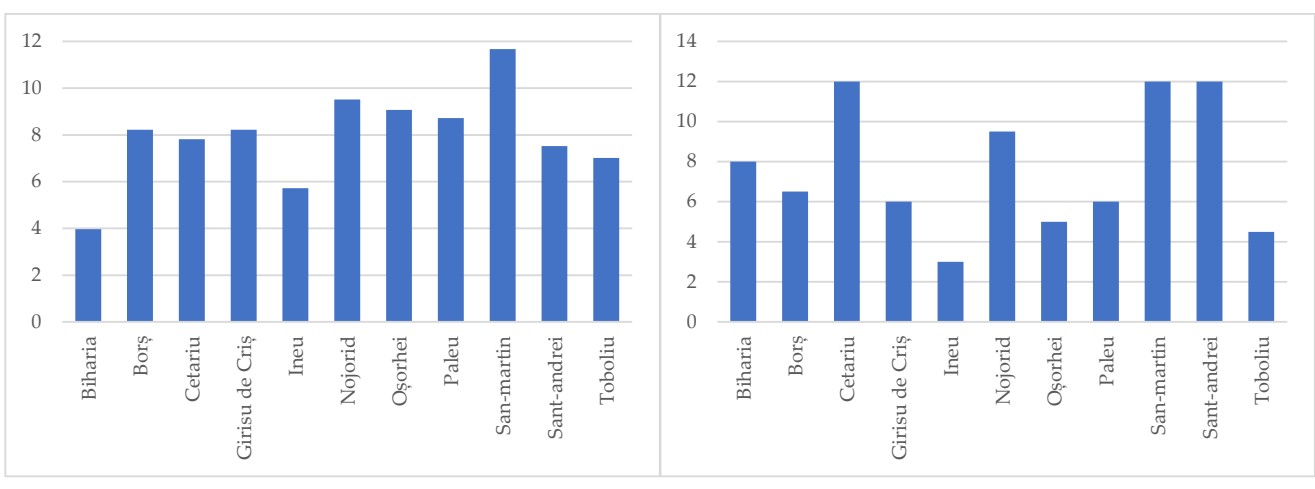

Natural tourist attractions                    Man-made tourist attractions

**Figure 2.** OMA communes natural and man-made attractions supply based on Tătar et al. (2018) and Tătar et al. (2021) [57,59].

By consulting the OMA local authorities, the Strategies of Sustainable Development 2021–2027, most communes tend to promote their most attractive natural resources and few set tourist-specific goals such as, for instance, to increase the occupancy level and length of stay as in the case of Sânmartin [47] commune, or to preserve the environment and to promote the local tourist attractions as in the case of Biharia and Ineu communes [61]. Besides Sânmartin, Biharia, and Ineu communes, which have some specific goals towards tourism, the rest of the communes which score very low actually only rotate around the same generic aims referring to tourism. Tourism is rather approached generically in most communes' strategic plans, except for communes where tourism demand actually occurs, as in the case of Sânmartin.

When referring to the man-made attraction concentration, the communes of Sântandrei, Sânmartin, and Cetariu are topping the list with 12 points each, namely 48%, whereas those of Toboliu and Ineu feature the lowest values. The average score for the rural metropolitan area at this category is 7.68 points, compared to the maximum value of 25 points, which is close to the medium value. Despite Ineu commune's low value, it holds good premises for sustainable ethnic tourism, as one of the commune strategy's [62] stated aims is to capitalize on its local traditions and customs, referring to folk festivals.

From a chart comparison of the two categories, i.e., natural and man-made attractions, it can be noted that, in terms of natural attractions, the rural metropolitan communes have a higher propensity and a more balanced territorial outspread, with the gaps not widening among the communes, versus the man-made attractions, where the distribution is less balanced and the gaps among the communes widen. Most man-made attractions are held by just a couple of communes, an indicator of the fact that nature finds its sustainable and balanced way across in a territory versus the built environment. Topography, vegetation, climate, and hydrography elements [4] encountered in the rural OMA metropolitan belt stand as tourist attractions in themselves and do not require major investments, as they are basically contemplated and not necessarily physically consumed, except for thermal waters. Nonetheless, indirect consumption of thermal waters from Sânmartin commune had a direct negative and irremediable effect on endemic species [63,64].

Most natural attractions are found in the north-eastern and south-eastern part of the rural OMA, which accounts for the fact that they border the piedmont hills of the Apuseni Mountains. By contrast, the man-made attractions are located in the western and south-western part of the metropolitan belt [58] where a past higher human mobility occurred due to its close vicinity to the Hungarian border; there was, formerly, a more open space for migration as it stretches over the Western Plain. It is a space for transborder cultural interferences, which yielded a cultural output such as churches, fairs, and handicrafts of different ethnicities. Even the years of documentary attestation of the villages form the western part (i.e., the year of 1214) go deeper into history versus its eastern counterpart.

Thus, by summing the values of the two categories we obtain a view of the tourist resources outspread in the territory. The first place is occupied by the Sânmartin commune with 47.35%, which places it within an average tourist potential rank, and at the opposite end, we have the Ineu commune with barely 17.4%, which translates into a very low potential. The rest of the communes feature a low potential with values between 23 and 40% (Figure 3, Table 1).

Within the second direction related to dysfunctionalities of the tourist and technical infrastructure/facilities, we encounter the highest gap within the commune ranking per all communes at the accommodation supply category, where the commune of Sânmartin juts out among the other communes with a value of 47.3% of the rural OMA accommodation supply. The rest of the communes are at the bottom line, a situation which triggers a very low rural metropolitan average of barely 2.73 points compared to this category's maximum score of 20 points. The high value obtained by Sânmartin of 18 points (Table 1) is due to the two thermal water spas in its territorial administrative area, namely the Băile Felix and 1 Mai, which have a wide range of tourist accommodation units, leisure-related (aquaparks) and cure facilities (health recovery hospitals, treatment centres). The two communes of

Girișu de Criș and Toboliu scored zero points as they do not possess any tourist infrastructure, whereas the other communes barely reached two points. Besides Sânmartin commune, which has a tourist-related history since 1583 with the first Italian traveller who discovered the healing powers of the thermal waters [49] and whose accommodation infrastructure was mostly made up of hotels, during the communist time it provided a standardized product, but it tended to rejuvenate with newer accommodation infrastructure such as guesthouses, agritourist guest houses, and motels from the 1990s onwards.

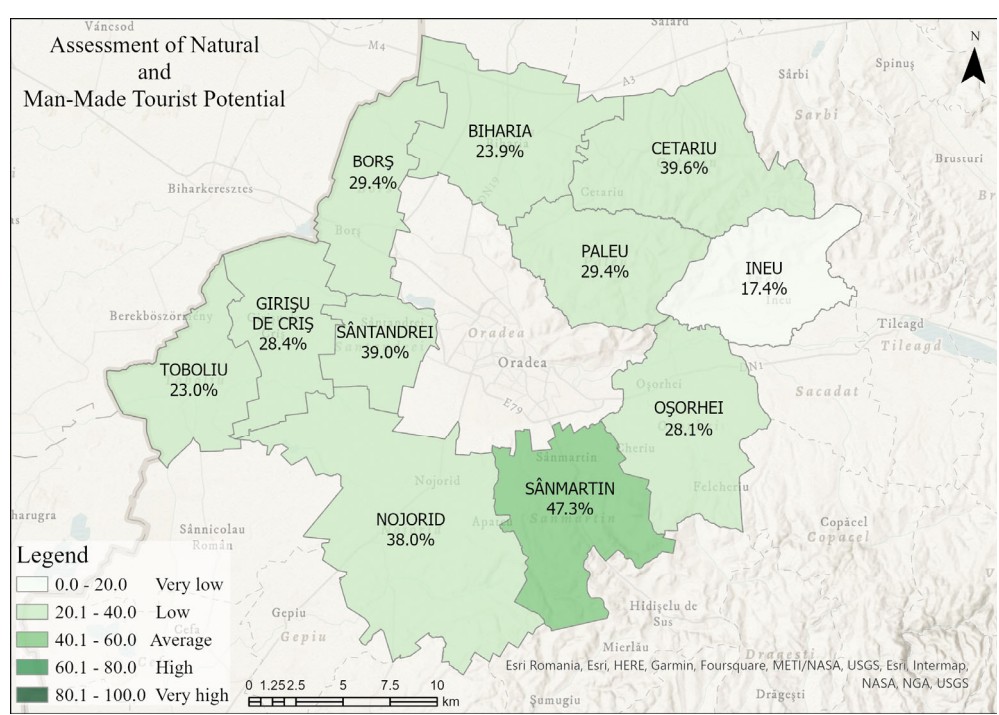

**Figure 3.** Rural OMA assessment of natural and man-made tourist potential. Source: own elaboration based on Tătar et al., (2018) and Tătar et al., (2021) [57,59].

The rest of the metropolitan communes are very poorly represented and are close to the bottom line, with most of their accommodation infrastructure having emerged after the 1990s in the form of guesthouses, agritourist guest houses, or other family-type accommodation units. After 2000, the diversification and accommodation infrastructure increase has been extremely dynamic; in 2008, there were merely 24 accommodation facilities, but in 2020, we can count 128, an increase of 533% [48]. Among them, the guesthouses skyrocketed from 9 structures in 2008 to 77 structures a decade later. In terms of good value for money, guesthouses are better rated versus hotels [47]. The occupancy rate at an average of 50% yearly is higher within hotels versus guesthouses, also due to the treatment of state-subsidized coupons provided by the National Public Pensions' House for health-related purposes and recovery based on thermal waters. The flow intensity coefficient is 3.1, and in full season in the month of August, demand exceeds supply, obviously indicating a need for the flow's reorientation for a higher sustainability.

During the 1970s and 1980s, the Romanian spa's socialist policy consisted of massive accommodation buildings, and this trend also applied in the case of the two spas of Sânmartin commune when the state invested massively. The state subsidized trips for the labour force, which stood out as a tool to legitimize a strict political communist regime and people's free movement right [65]. On this backdrop, the tourist product was standardized in the sense that all accommodation units looked similar, with high-capacity hotels and with the same entertainment opportunities for all in the spas; the only out-of-the-ordinary accommodation units were the villas, as a remanent of the past. Tourists were treated as a mass and not according to individual preferences. This situation has changed dramatically

since the 1990s and 2000s, with new types of accommodation structures such as hostels, agritourist guesthouses, and bungalows. In spite of the lower number of beds versus hotels, it is better adjusted to the new contemporary tourist's needs and preferences, and is targeting a niche market [66] rather than mass tourism, which the commune of Sânmartin abounds in recently. The tourist pressure over Sânmartin has yielded negative results for the current wildlife, leading to an in situ extinction of endemic fauna and flora [55,56].

The new rearrangement of communes surrounding the core city of Oradea into a metropolitan area now sees Sânmartin commune with its two spas as Oradea city's main tourist satellite with a huge accommodation capacity of 8656 beds in 2020, exceeding by far that of the city of Oradea, which had 2672 beds in 2020 [67]. Therefore, tourist flows are mostly drawn by Sânmartin commune in disfavour of the other metropolitan communes, which are almost deprived of tourism activity when comparing the huge gap between Sânmartin and the rest of the communes. Still, the low accommodation representation within the rest of the communes can become a strength in the sense of being more sustainable and producing less strain on the local environment. This could be an opportunity to be capitalized upon by local authorities for a more balanced metropolitan tourist flows reorientation.

In the technical facilities supply, scores are more balanced. Borș and Nojorid communes rank first (Figure 4), which benefit from a favourable position due to the existence of superior-ranked communication routes, such as European national roads and the existence of Oradea International Airport in Nojorid commune. In addition, Borș is a border commune and also a border point between Romania and Hungary.

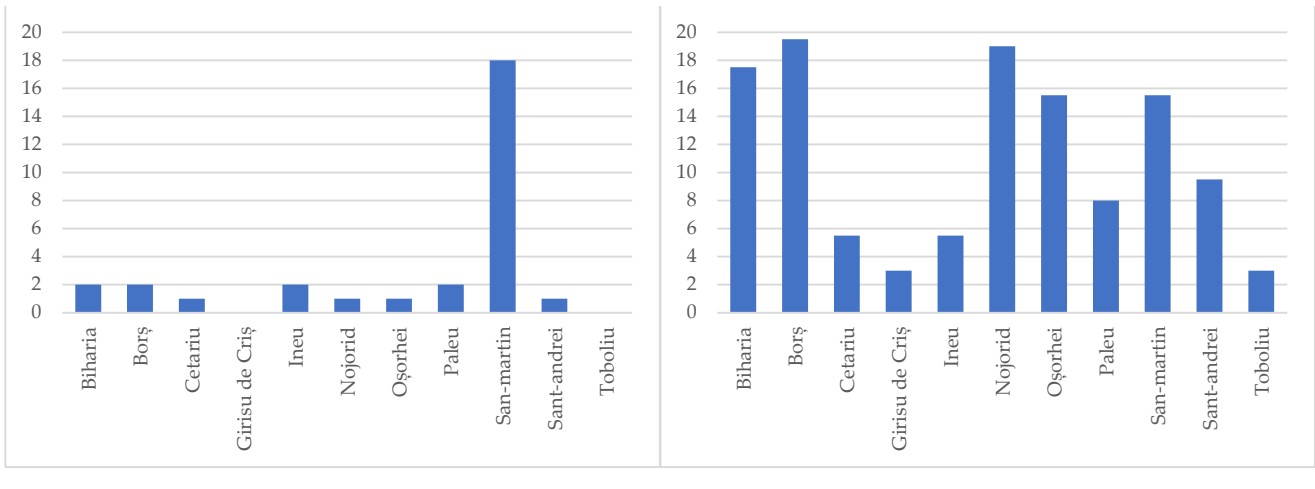

**Figure 4.** Tourist and technical facilities supply. Source: own elaboration based on Stașac et al. (2020) [48].

At medium potential (between 50 and 60%), we find the communes of Biharia, Oșorhei, and Sânmartin, which are crossed by European national roads. Due to their peripheral location compared to the main communication routes, the communes of Cetariu, Ineu, Girișu de Criș, and Toboliu, with 10–20% (Figure 4), are disadvantaged from this point of view. The conclusion can be drawn that the more higher-ranked communication routes are passing through the commune, the better they score at the technical endowment category.

By summing up the scores of the two categories above, we obtain some territorial dysfunctionalities. Therefore, the communes which score very low and are in deficit at both tourist and technical supply are Toboliu, Girișu de Criș, Cetariu, and Ineu (ranging between 5 and 20%), followed by Nojorid and Borș with an average potential. Sânmartin commune tops all with the highest potential at both categories of over 60% (Figure 5). Sânmartin commune scored very high because it hosts the two internationally reputed thermal spas of Băile Felix and Băile 1 Mai, therefore it has the highest density and diversity of all OMA metropolitan in terms of accommodation supply, nonetheless, in terms of technical

endowment, Sânmartin commune lost ground versus Bors commune, as the former is not yet linked to the gas network. Borș commune, on the other hand, scores very well at all analysed items of the technical facilities supply, being equipped with gas, sewage, and access to European and national roads. Location is a territorial-related strength, as it is a border commune with Hungary and witnesses a high international border transit from this perspective. Despite a high border crossing transit, Bors commune remains just a transit commune, with barely two accommodation units in the form of a hotel and a guest house, whereas Sânmartin commune has the highest rural OMA accommodation supply with 237 diverse accommodation units.

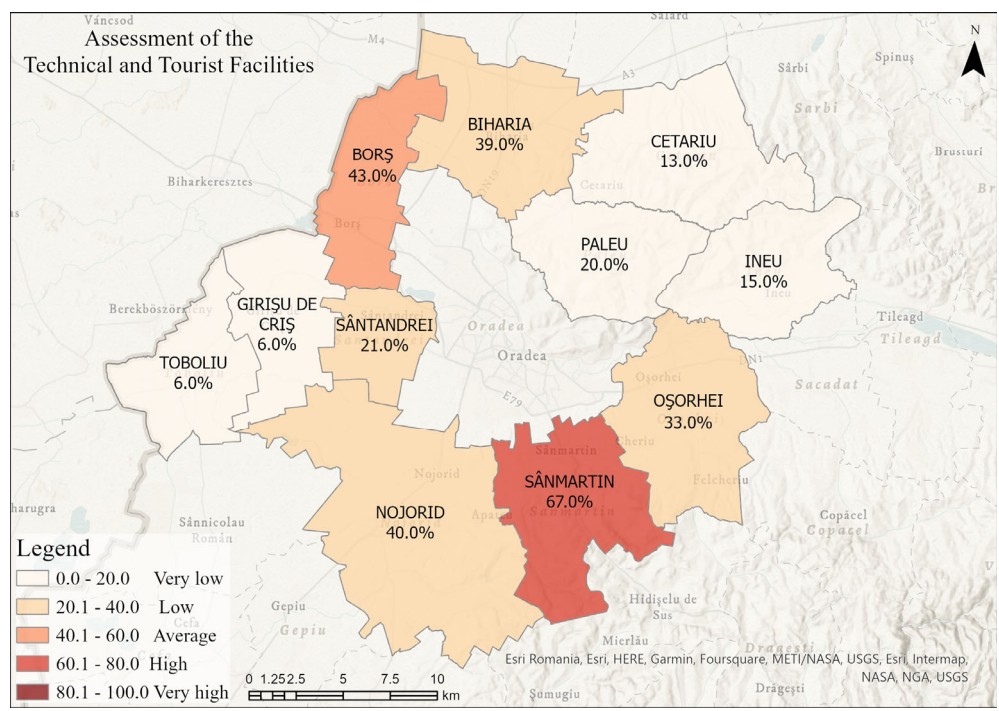

**Figure 5.** Rural OMA assessment of technical and tourist facilities potential. Source: own elaboration based on Stașac et al. (2020) [48].

By accomplishing the total rural OMA assessment of the natural, man-made tourist attractions, and technical and tourist facilities, we supply a commune ranking result where we find three communes with a very low score, below 20%, such as Toboliu, Ineu, and Girișu de Criș commune; seven communes with a low tourist potential between 20 and 40%, such as Paleu, Cetariu, Sântandrei, Oșorhei, Biharia, Borș, and Nojorid; and the single commune of Sânmartin with an average tourist potential of over 50% (Figure 6).

The study indicates that from all eleven communes, the commune of Sânmartin has the highest tourist potential of all OMA communes. A few other OMA constituent communes also hold good values close to the average, such as Nojorid and Bors, but they are still at the beginning of their tourist activity. Location and type of communication infrastructure are key for their higher score, namely the former hosts the International Airport of Oradea and the latter contains the most used border crossing point with Hungary, i.e., Varsand. This closeness also triggered many crossborder projects, implementation which further stimulated tourist demand on the metropolitan area. A few implemented projects from 2021 are: ROHU 265 "Let's celebrate our traditions together" within the Interreg V-A program Romania–Hungary; CYCLEWALK project funded by the program INTERREG Europe; ROHU29 "Conservation and protection of ecosystems endangered by lack of thermal and freshwater in crossborder area", within the Interreg V-A Romania–Hungary; ROHU 200 "Crossborder events for crossborder citizens" within the Interreg V-A Romania–Hungary; and ROHU 319 "Joint Program For Youth Cross-Border Cooperation", within the Interreg V-A Romania–Hungary [68].

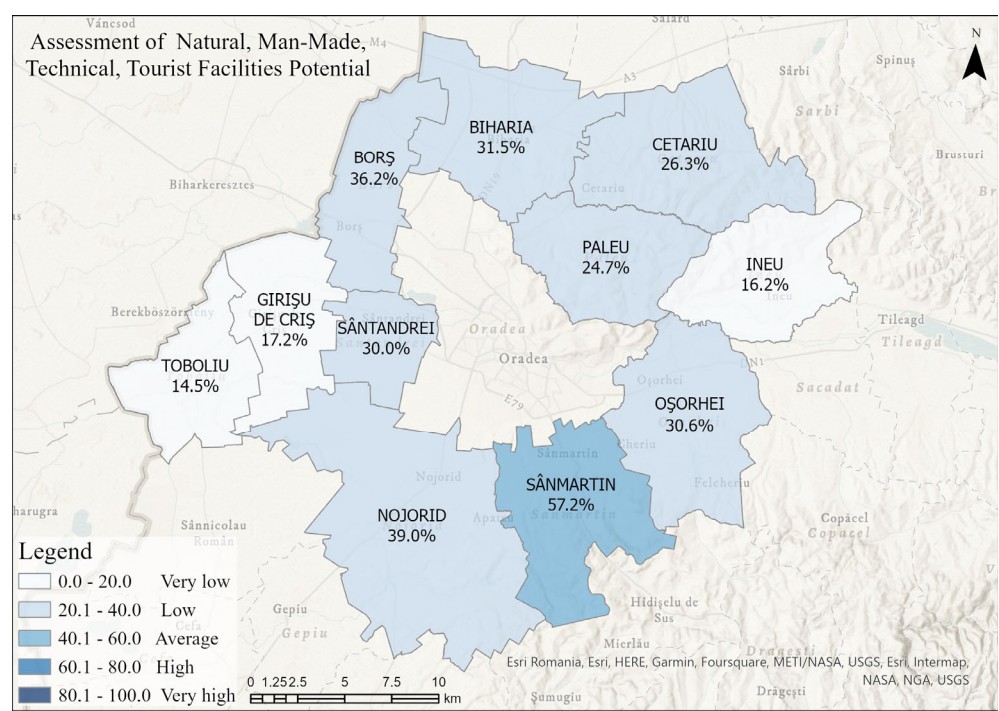

**Figure 6.** Rural OMA assessment of natural, man-made, technical and tourist facilities potential. Source: own elaboration based on Staşac et al. (2020), Tătar et al. (2018) and Tătar et al. (2021) [48,57,59].

The eleven OMA communes' natural and man-made heritage assessment is an important step in deciding the priorities into capitalizing and marketing for tourism in this area. Furthermore, it can help the elaboration of policy planning strategies on pondering whether the medium to high tourist potential commune need further development or if the approach should be to grow interest and development in less significant and low tourism potential rural areas [69]. Through the assessment of tourist resources and facilities, local governments can relocate funds [22] for the low potential tourism communes which can bring about a balance for the peripheral communes, especially, and thus a more sustainable consumption throughout the entire rural OMA. It is obvious that the higher potential OMA commune can better cater to the demanding tourists needs, but the lower tourist potential communes can complement the OMA tourist by offering authentic experiences [70] given by the traditional lifestyle or ethnic communities as proposed for the communes belonging to the marginal community tourism models. The study limitations reside in the proposed model's low replicability for metropolitan areas, as they are not conceived after a mathematical model but rather by utilizing the item ratings of Table 1. Nonetheless the current study's contribution is that it provides an auditing of the tourist potential of each OMA commune, which highlights their competitiveness and hierarchization [71].

## 5. Tourism Models Proposals

The exploratory and synthetic analysis of the dominants factors, determinants, and variables [72] that decisively influence current and potential tourism activities has allowed the configuration of three models of tourism in the OMA (Figure 7). The applied methodology for the potential evaluation further on allowed the elaboration of three themed models relying on the statistical results for the natural, man-made attractions, and tourist facilities. The motivation of the first model (A) stems in the statistical analysis scores related to cure factors, accommodation, and cure facilities, which applies to communes such as Sânmartin and Nojorid. For the second model (B), the statistical items scored were attractions such as nature tourist reserves, folk craft, and tradition which target communes as Cetariu, Paleu, and Biharia. For the third model (C), the statistical items scored refer to the natural resources in association with agricultural activities and targeted communes such as Girişu

de Criș, Toboliu, and Borș. The models represent the authors' own elaborations and stand as potential recommendations for the policy makers of the OMA communes rural strategies, as well as for researchers as a means to bridge the gap in the current OMA tourism literature approaches.

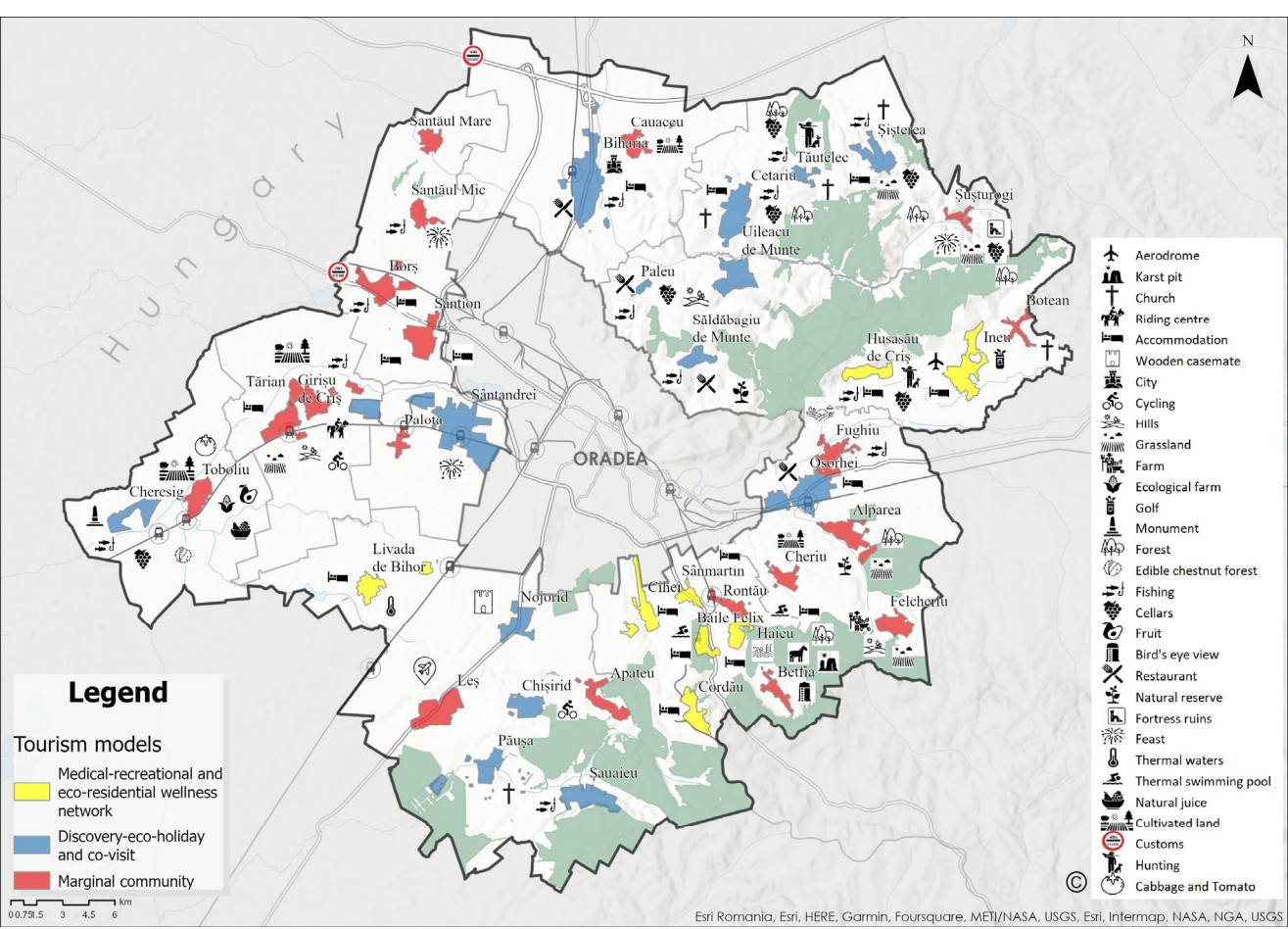

**Figure 7.** The situation of tourism resources and their thematic activities, and the three resulting tourism models in OMA villages. The polarizing effect of Oradea and the systemic partnership between the city and the OMA villages should be noted, through the generated set of socio-economic and ecological relationships.

**A. The medical-recreational and eco-residential wellness network tourism model.** Wellness is a set of activities and guidelines for individuals who pursue a healthy, active, but also relaxing lifestyle, in other words, willingness to travel and willingness to treat [73,74]. This type of tourism has unique features at the level of the OMA, because as a socio-economic and ecological system, with technological support and sustainability orientation [48,75–78], it highlights a dominant tourist function, but also a mix between complementarity and dependency. The dominant tourism function of medical–recreational wellness is decisively marked by the facilities and medical infrastructure for health, maintenance, physical, and socio-emotional well-being programs in the spa resorts of Băile Felix and Băile 1 Mai (Figure 8). In addition, the medical-hotel facilities provide modern equipment and procedures for all ages (but especially anti-aging treatments for the large number of elderly Israeli and German tourists) and indoor spa infrastructure, the outdoor one operating regardless of the season. It is the positive effect generated by one of the natural tourist resources, namely the thermal waters, which set the tone for high tourist demand and create sustainable economic premises. Including the entertainment side, good mood and satisfaction are based on an array of swimming pools, water parks and lakes

planned within a thermal species landscape (i.e., water lilies, turtles, tropical fish), including the endemic fish on the Pețea River with *Scardinius racovitzai* that use the same thermal water with certified sanogenic properties. Moreover, the economic–territorial system with the spa resources of Băile Felix–Băile 1 Mai is a consistent provider of jobs, registering in 2000 a number of 963 employees, with 56.3% employed in tourism out of the total number of employees, maintaining around 55% including in 2016 [79].

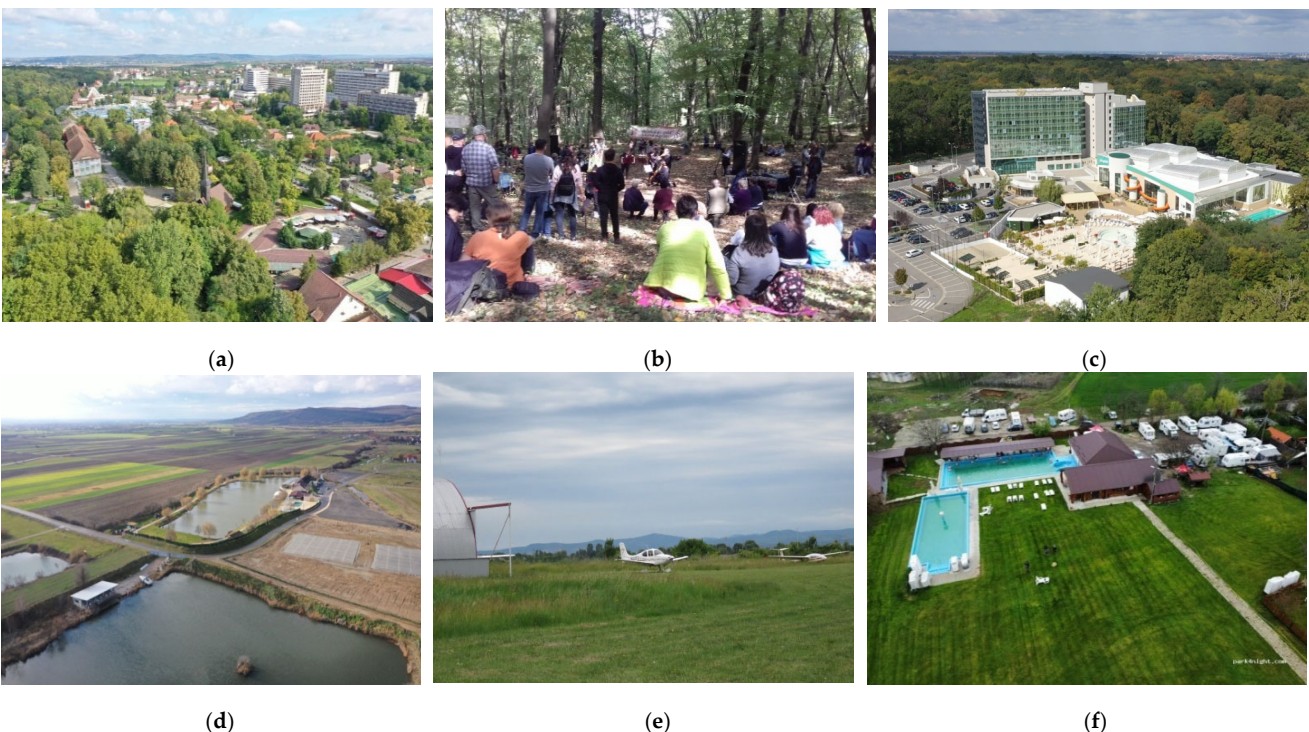

(**a**)    (**b**)    (**c**)

(**d**)    (**e**)    (**f**)

**Figure 8.** Examples of activities and valorisation of the tourist space that fit the medical–recreational and eco–residential wellness network tourism model. (**a**) The Felix Baths, the balneomedical and recreational infrastructure and the accommodation capacities that are spread over a generous area, competing with the hilly and forest surroundings. (**b**) A happy complementarity between a generous area occupied by an ecosystem of mature hornbeam and linden forest and artistic and ecological cultural initiatives that attract a significant number of tourists staying in Băile Felix Spa and Băile 1 Mai Spa; (**c**). Tourist space is maximized in a medical-hotel unit. (**d**) Camelot Resort in Husasău de Criș: an example of competition for space, but also of functional complementarity with the surrounding natural environment. (**e**) The favourable morpho-biogeographical conditions of nature in Ineu offer excellent recreational opportunities also for Romanian and foreign tourists, who combine the pleasure of walks in the surroundings with those involving a more adventurous character, namely gliding and light aircraft flying. (**f**) The specific plain lands offer tourism investors in Livada de Bihor the ability to put in tandem accommodation services, including space for caravans, enjoyment in thermal pools and restaurants, and the discovery of typical rural life nearby.

Complementarity and dependency are engaged by the satellite localities of these two resorts (i.e., the localities of Sânmartin, Cordău, Haieu, and Cihei). They enter the network of main tourist service providers of Băile Felix–Băile 1 Mai, mainly for accommodation and bed and breakfast through several hundred small hotel units, tourist guesthouses, and private properties. The ecological and topographical conditions are favorable, without massive occurrence or dangerous frequency of climatic, hydrological, and geomorphological risks. The local nature, configured by a hilly relief with a moderate slope and a valley corridor, with a mild temperate topoclimate, creates optimal conditions for silvo-steppe, with heather, oak, and hilly plateau meadows (i.e., Băile Felix Forest and surroundings). Here you can find paths and running tracks preserved in their natural state, which facilitate

walking and jogging. The second pole of the wellness network tourism is also located on the eastern side, where the couple of localities Husăsău de Criș-Ineu make the water, both cold water lakes and thermal water pools, the object for practicing recreational wellness activities such as swimming, sports activities such as fishing, archery, parachuting, gliding, horse riding, and golf; light leisure activities such as walks and discovery of the aquatic and meadow nature; and cultural activities related to festivals and gastronomy. An unusual presence in this tourism model is Livada de Bihor Village on the southern side. In a plain area, an island of thermal water use appears in the form of a thermal water pool park, including accommodation facilities in the shape of small elegant wooden cottages on the background of a peaceful atmosphere and provision of quality services for visitors.

At the level of each of the two poles of this tourism model, a spatial and resource dispute, as well as a very good capacity for investment and renovation of tourist units, can be noticed. In the case of the Băile Felix–Băile 1 Mai pole, the territorial advance is made on the north-west–south-east axis, where the capacity for investment and renovation of tourist facilities is in the detriment of the pastures of the neighboring villages and partially of the pine forest (Figure 8). There is also a sense of vigor in the interest with which new personal housing structures and guesthouses in Sânmartin and Haieu are attached to the large medical–recreational tourist pole, highlighting the eco-residential character. It is about choosing to invest and build such tourist accommodation and catering structures, as an outcome of their location near the Băile Felix Forest, but also by eliminating some steppe areas. The second pole is that of Husăsău de Criș-Ineu, which claims the investment effort through the space competition to the detriment of the northern part of the Crișului Repede meadow (Figure 8).

In the cases of both poles, we can speak of an encouraging, constant level of technological, urban, and recreational innovation and renewal. For Băile Felix–Băile 1 Mai, we talking about not only state-of-the-art physiotherapy and kinesiotherapy equipment, but also an ambitious urban regeneration and reconfiguration program with EU funding (i.e., pedestrian walkways with mineral materials and solutions for a modern design), reconfigured parking lots, ecological light sources, and small landscaping such as alignment and green cells, following a new vision that values the opening of spaces for walks and comfort for small groups. For the Husăsău de Criș-Ineu pole, we are dealing with innovation within a short time frame of 1–2 years.

This model of tourist practice is characterized by a good motivation for mobility, both at the individual and group level. On an individual level, we are talking about encouraging walks through forests and meadows, with boats on lakes, as long as there are paths and the difficulty of the trails is generally easy to medium. For the group level, we are talking about round trip mobility in these tourist areas. This mobility is facilitated by the nearby existence (3–6 km) of the E60 road for the Husăsău de Criș-Ineu pole, and for the Băile Felix–Băile 1 Mai pole and neighboring towns, the Oradea Train Station is important, as well as the Oradea-Băile Felix train trip (21 km) and theOradea-Băile Felix road near Băile 1 Mai (6–7 km). Oradea International Airport is added, located approximately 11 km away, with 5 medium courier services and many charters, which facilitate the rapid transfer of tourists, especially foreigners.

**B. The discovery-holiday and co-visit tourism model**. It sheds light on an interaction tourism that generates two situations. One is where tourists interact with natural and man-made tourist attractions in the OMA. The second one refers to a new condition of a part of the tourists after the contact with the geographical environment they visit: they return on other occasions and with other visitors to explore other spaces, other tourist resources, and other trails. In addition, in the case of the OMA, this tourism model highlights another (positive) impact of the visitor's interaction and socialization with the local tourist space. Attracted by the personality of the landscapes that exude calmness and safety and which incites curiosity (Figure 9), some of the visitors make the decision to purchase land on which they build a holiday home or directly buy a property located in an attractive landscape area or in the proximity of a local tourist attraction (Figure 9). Moreover, in this case,

we can speak of a partnership between the tourist/visitor and local nature, in this way developing tourism from the micro-scale to a tourist space that becomes larger and more potent relationally and functionally.

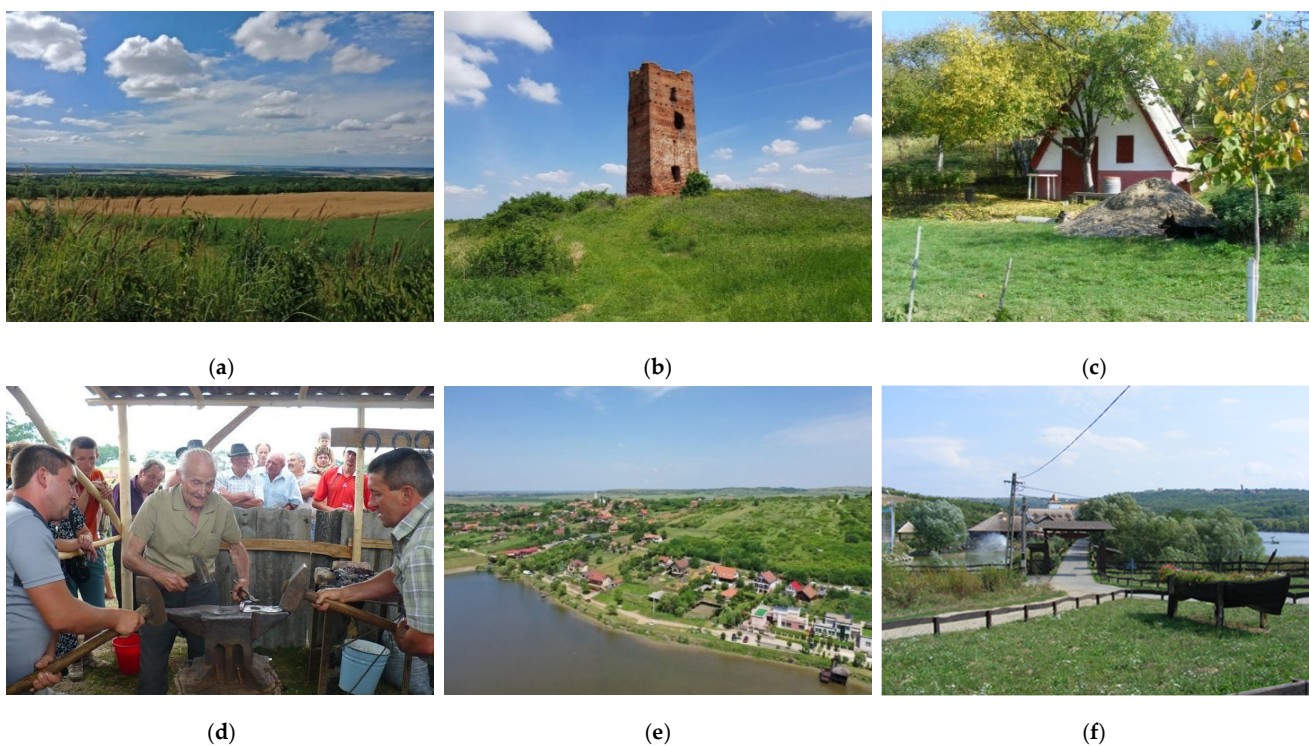

**Figure 9.** (**a**) Very gentle and very wide slopes between Șișterea and Șușturogi: cereal fields, meadows and forests waiting to be discovered by nature lovers; (**b**) The Cheresig Donjon: between space, antiquity and history lovers; (**c**) Holiday home in Tăutelec; (**d**) The village traditions and crafts attract tourists of all ages and professions. Picture from Cetariu Days; (**e**) In the foreground, in the new part of Paleu Village, we notice a dispute between the new habitat (holiday homes, restaurants and guesthouses) and the two environments (the lake and the hill occupied by pastures and shrub formations), with the result that discovery and weekend tourism and leisure set the tone for the sufficiently pronounced artificialization of this part of the village; (**f**) The location in the immediate vicinity of Oradea makes the village of Săldăbagiu de Munte recognize the fierce dispute and competition for land of the city, with one of the results being the installation of the tourist complex Hanul Pescarilor.

The simple scheme of this model, according to which tourism is performed in the OMA, is as follows: you want to discover new places; you choose (influenced by social networks or other sources); you decide the destination; you engage in physical tourist activities; you look for satisfaction and you estimate to come back; you want to enjoy more of the new destination and the tourist attractions; you buy/rent a property or build a holiday home to enjoy all the modern amenities and be close to your favorite attractions. At the village level, this type of tourism model is concentrated mostly in the northern and southern parts (Figure 9), where the hills and forests dominate. A less consistent prevalence of this tourism model is also found in the villages in the western and the central-western part of the OMA, the explanation being that the villages and their related territories are located in the plain area, where the competitiveness of the tourist attractions and the interactions of visitors with the geographical space are weaker.

In an overwhelming majority, the tourist attractions are independent of each other, but the same geographical space, sufficiently energetic from a geomorphological point of view, is permissive for the establishment of tourist itineraries and engaging groups of a few people or single tourists for ecotourism activities [80]. These are routes of medium difficulty

in the form of a network of briefly arranged and oriented paths and trails, between 4 and 8 km or even 24 and 25 km long. They merge historical–religious and cultural heritage (i.e., old churches, cemeteries, cellars on the hillside, ruins of earthen fortresses, and thematic village festivals—Figure 9) with the natural heritage (i.e., nature reserves, natural meadows, lakes, mature cypress forests, holm oaks and edible chestnuts, and built countryside typical of the Hungarian minority). An example of such a trail connects Paleu-Săldăbagiu de Munte-Uileacu de Munte-Cetariu-Tăutelec-Sișterea. Due to such a rich distribution of local attractions, the place attachment occurs not only for the residents of Oradea, but also for tourists, who become attached to the place and will return, some even building own holiday homes. For the southern part, a themed itinerary is the cycle route of Oradea-Păușa-Chișirid. The level of difficulty is reasonable and can train a group of cyclists up to 10–15 people. Thus, Ilieș et al. (2013) proposed 11 cycling itineraries for OMA with a total length of 296.5 km [81]. The discovery–holiday fusion is also valid for the cycling itinerary of Oradea-Sântandrei-Tărian-Girișu de Criș, where cyclists enjoy watching the river course, the meadow, the hydro-hygrophilous vegetation, and the widening of the Crișul Repede River water course, or even stop for a fishing and bird watching session.

Moreover, a dependency is created both for visitors from Oradea and for very active tourists, eager for sports and nature. Romanians and foreigners who come from Băile Felix spa can rent bicycles or other means of electric mobility. Moreover, the mobility and communication of these tourists are of a good quality. There is a network of antennas for GSM communications that are sufficiently well positioned spatially and spreading mostly over the hilly interfluves that are most pronounced altimetrically. The distances between these antennas in this area are of a maximum a few kilometers in a straight line, and the geolocation through GPS coordinates is equally good in quality. The mobility of tourists is also ensured by a network of public and private buses departing from Oradea (from two bus stations), but the frequency is not the most convenient for tourism (1 departure/2 h). Only coaches and minibuses are good substitutes for buses, which can be rented from various travel agencies.

**C. The marginal community tourism model**. Upon a brief examination of the semantics and thematic content of this tourism model, it addresses a space occupied by poor people or who manifest a chronic lack of education and professional training. Beyond the delicate acceptance that there is a minimum of truth in these assertions (i.e., a part of the poor inhabitants of the villages, seasonal agricultural workers poorly paid for agricultural or forestry work, and disadvantaged communities of rural Roma people), we must accept that it is about a certain tourism demand that relates to the determining geographical space and less to the population variable.

In the case of the villages that are subject to this tourism model, we must accept that the space determinant makes a rough reference to the dominant geographical position of the villages: the overwhelming majority are located on the edge of the OMA, at a relatively large distance (even 8–13 km) from the polarizing urban center of Oradea (Figure 10). Here, the physical distance is no longer limited for tourism and tourist activities. In this case, it holds special features. First of all, it is about the dependence of some tourists on the distance, the flirtation with the idea of "searching so that what I am looking for is as far away as possible" (the case of the villages of Șușturogi, Botean, Leș, and Toboliu). Not even the determinants of physical or ecological conditions, the level of innovation, or the investment in tourism programs fully account for the option for tourism here, as innovation and investment are sometimes reduced or non-existent. We are even talking about an antagonism, in that individuals who practice domestic tourism are interested in taking part in tourist activity without intersecting with other individuals. In the case of foreign tourists, the fact of being in these villages and in rural areas far from Oradea is attributed to curiosity, not necessarily seeking solitude. They seek pure nature and unspoiled village culture, not even bothered that they cannot find museum networks or that they are at a great distance from a hotel or guesthouse. Weekend tourism (i.e., picnic and barbeque, picking berries from the forests and pastures of Betfia, Apateu, and Șușturogi villages) and

event tourism (i.e., communal days and seasonal events from Sântion, Leş, and Toboliu villages) is what satisfies domestic tourism. In addition, for foreign tourists, hiking (both in the villages in the hilly east and in the plain west, with swamps, ponds, and meadows), gastronomic tourism (i.e., Toboliu cabbage, Alparea cheeses, hot pepper, and cherry juices of Apateu) are attractive for foreign tourists, as well as tourism for photography and landscapes (Figure 10).

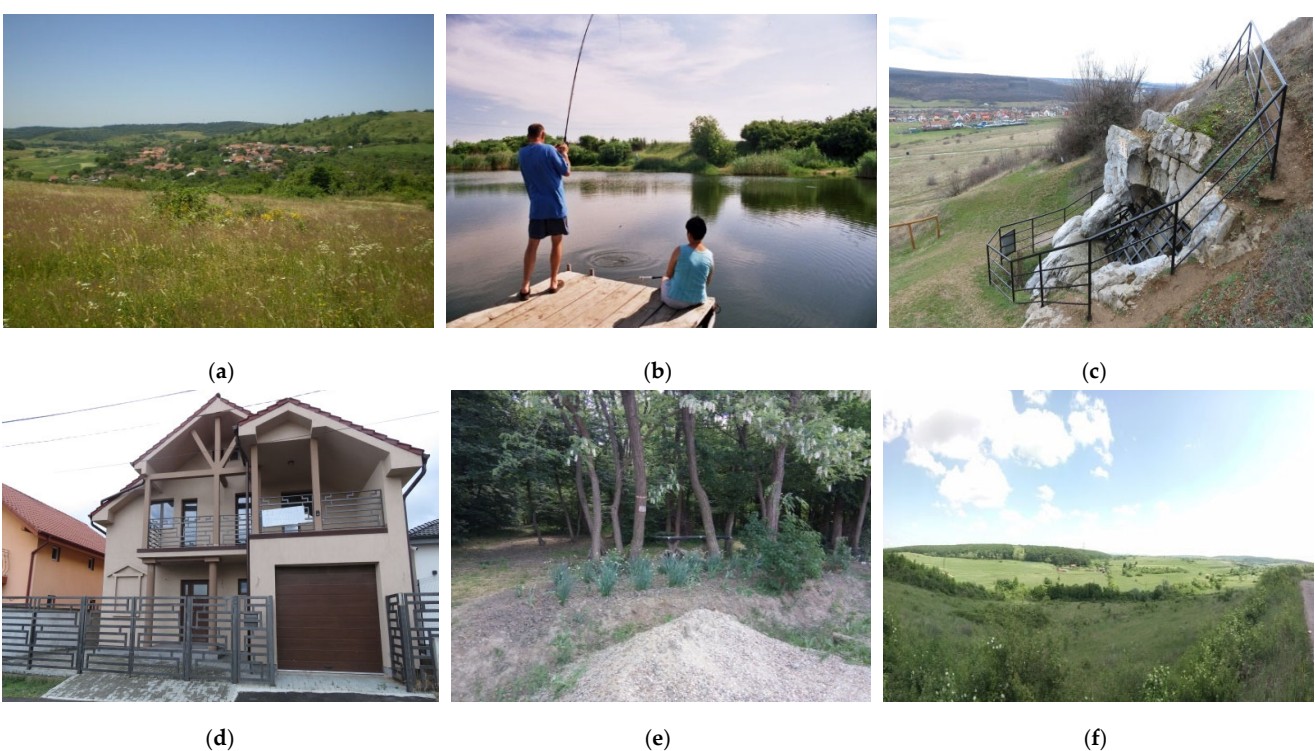

**Figure 10.** (**a**) Idyllic rural landscapes at Șușturogi invite you to hike and discover the traditional houses belonging to the isolated Romanian community; (**b**) The Santău Mic lakes are attractive not only for locals and citizens of Oradea but also for foreigners, given the location of the lakes near the Hungarian border; (**c**) A geotouristic resource such as the Betfia Aven creates, beside touristic interest, also conflict with fans of motorized touristic activities; (**d**) A property that once served tourists awaits its new owners who may even be tourists; (**e**) The Nature Reserve Narcissus Forest near Alparea is of interest mainly for scientific and ecotourism; (**f**) For hiking and discovering rural nature, the wide hilly landscapes with meadows, deciduous forest and farms (including organic) of Felcheriu are easy to visit.

Public food centers are numerically reduced to an average of 1–3 units/village, being placed anywhere in the village.

This tourism model is a physical one and tourism is influenced by social networks (almost all communes have their own website), the few guesthouses, holiday homes that are rented out, and entrepreneurs for small businesses having a website on popular social networks. The possibilities of traveling by car or minibus allows a good mobility for tourists as a result of a well-maintained road network, despite the distance from Oradea.

## 6. Conclusions

Taking into account the fact that OMA spreads over a plain and low hilly relief where the natural tourist resources are even so reduced, we deem that the score obtained by most of the communes is a reasonable one and is backed by the OMA location in the Hungarian crossborder area. For the metropolitan area, thermal waters are the main impetus for visitation, for leisure and cure tourism.

In terms of the tourist concentration poles of both natural and man-made rural OMA attractions, they are preponderant in the west in Sântandrei, south in Nojorid, and in southeast in Sânmartin communes, as well as to the north-east in Cetariu commune. Sanmartin emerges as the hotspot in terms of visitor attractions concentration with the highest value. The best tourist potential is held by Sânmartin commune for all the analyzed categories, such as natural and man-made attractions and technical and tourist facilities supply, which features an average potential, followed by Nojorid, Borș, Biharia, Oșorhei, Sântandrei, Cetariu, and Paleu with a low tourist potential, and Girișu de Criș, Ineu and Toboliu communes with a very low tourist potential (Figure 6). This almost tourist-exclusive consumption in a proportion of over 93% [59], which is currently occurring in Sânmartin commune, shows that flows need to be dispersed more evenly among other metropolitan communes, and the current consumption is unsustainable. The other single commune where thermal waters are capitalized is Nojorid, which hosts an operational swimming pool park which also uses thermal water for bathing and cure purposes, but the demand gap between Sânmartin and Nojorid is still very wide; the latter registered barely 5153 arrivals during 2001–2020, whereas the former registered 2,791,492 arrivals during the same time interval [59]. If the accommodation facilities and bathing pools infrastructure were increased and upgraded in Nojorid commune, it could divert tourist over redundant flows of Sânmartin and reduce anthropic pressure on the southern metropolitan part, and thus yield a more sustainable consumption. The touristic pressure on Sânmartin commune has yielded negative outcomes for its natural habitat, mainly referring to the natural reserve of Băile–1 Mai spa from where three thermal endemic species have become extinct [63,64].

The constituent rural OMA communes display a varied range of tourist supplies such as avens, churches, riding centres, nature reserves, thermal waters, wine cellars, festivals, farms, fishing ponds, and golf courses (Figure 7), unevenly spatially dispersed, addressing certain segments of tourists such as those interested in cure, eco-discovery, and ethnic communities. Not all communes have the same types of attractions, but they are complementary.

The technical and tourist facilities are disproportionately placed throughout the rural OMA, and it shows that the tourist infrastructure is at very low levels in all analysed communes except for Sânmartin, whereas the technical facilities supply is quite balanced throughout the rural metropolitan area. Therefore, it is a technically well-endowed rural metropolitan area with the only tourist supply hotspot in Sânmartin commune, where most tourist consumption and production occurs, which engenders unsustainable practices.

For a sustainable OMA tourist consumption, the analysis of all factors, determinants and variables led to the identification of three tourism models in the OMA, such as the medical–recreational and eco–residential wellness network tourism model, which relates to thermal waters and applies to localities such as Băile Felix and 1 Mai spa resorts, Sânmartin, Cordău, Haieu, and Cihei as tourist services providers, with Husasău de Criș, Ineu and Livada de Bihor related to the same wellness model. The discovery–holiday and co-visit tourism is the second model that applies to the southern and northern part, where hills and forests dominate and are suitable for walking and cycling. The third model refers to the marginal community model and applies to the marginal OMA localities, such as Șușturogi, Botean, Leș, and Toboliu, for a full immersion into an unspoilt bucolic life.

The study results can be used by the political actors for the reconfiguration of tourism policies, as well as by the private and local investors with a view of developing further tourism projects, and they can be transferred and adjusted to different spatial scales according to their specificity. The OMA is a very dynamic territory which has implemented many European projects, and highlighting the rural OMA tourist potential can help investors to catch on to the tourism development rising trends and opportunities, even for the smaller communes such as that of Bors. The latter applied and was granted the status of a local interest tourist resort in 2020 for the purpose of absorbing European funds and attracting investors [82].

**Author Contributions:** Conceptualization, C.-F.T.; methodology, R.L.; software, L.B.; data curation, M.I.S.; investigation, I.D., S.N., M.S.S.; original draft preparation, C.-F.T., I.D.; review and editing, I.D., R.L. All authors have read and agreed to the published version of the manuscript.

**Funding:** This research was funded by the University of Oradea.

**Data Availability Statement:** Data available in a publicly accessible repository. The data presented in this study are openly available in repository *Sustainability*, Special Issue "The Influence of Space on Tourist Activity: Supply, Demand, Competitiveness, Sustainability and Innovation".

**Acknowledgments:** The authors are grateful and would like to thank the academic editors and to anonymous reviewers for their valuable comments on the manuscript.

**Conflicts of Interest:** The authors declare no conflict of interest.

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
