# Peer review of "Oradea Metropolitan Area as a Space of Interspecific Relations Triggered by Physical and Potential Tourist Activities"

_sustainability, doi:10.3390/su15043136_

Round 1
Reviewer 1 Report
Dear Authors,
The manuscript presents a quite interesting study, but they are some aspects to be clarified:
- I the first part maybe it will be better if the authors can proper define the aim /purpose of the paper
- The research result part is to short, this part is the main part of the research!!! The interpretation of the results must be put in this part, ….in my opinion at least.
What are your future proposals for the area studied, taking in consideration the conclusion from this study?
Author Response
Dear Authors,
The manuscript presents a quite interesting study, but they are some aspects to be clarified:
- I the first part maybe it will be better if the authors can proper define the aim /purpose of the paper
- The aim of the paper has been defined at the end of the introduction and in the abstract
- The research result part is to short, this part is the main part of the research!!! The interpretation of the results must be put in this part, ….in my opinion at least.
- According to the suggestion we have merged the chapters 3 and 4 into a single one under the heading Current Research Results and Discussions
What are your future proposals for the area studied, taking in consideration the conclusion from this study?
- From the analysis of the literature and of the commune’s development strategies we identified a gap for the tourism approach and we will propose to upgrade these strategies with substantial tourism information outlined by the study results
- Further research proposals relate to to the tourism impact on the commune’s host society, the impact of tourism on rural OMA demographic resources relating marginal villages and their revitalisation through tourism.

Reviewer 2 Report
This is an interesting and well-prepared paper, which can be read by a wider audience as it connects destination management, spatial planning and sustainable development. The overall quality of the paper is very good, still some shortcomings should be addressed before publication. The authors are advised to address the following issues:
1. The paper is primarily based on the “Methodology of the 25th of April 184 2016 for the Analysis of a Territory’s Tourist Potential”. This context and content is not explained and since this is probably a local policy text it would be necessary to have some more information. It is suggested to at least briefly present its main points, especially for the readers that are not familiar with the planning regime in Romania.
2. The presentation of the methodology of the survey can be improved, the reader doesn’t understand whether hypotheses have been made and have been met, the key questions should be highlighted in a stronger way.
3. Part 5 of the article is far too extended (from page 11 till 24) and should be shortened: some pictures could be deleted, and some information merged so that the reader can detect the key points.
4. The most important weakness of the paper is detected in part 6: the final discussion adds new information and is not reflecting on the initial research questions. To some extent it is also misleading as it doesn’t’ refer to the main points of the paper. My suggestion would to rewrite this part and focus on the take away messages from applying this method, include policy suggestions and propose further research if necessary.
Hope the above is clear and helpful, wishing you all the best.
Author Response
This is an interesting and well-prepared paper, which can be read by a wider audience as it connects destination management, spatial planning and sustainable development. The overall quality of the paper is very good, still some shortcomings should be addressed before publication. The authors are advised to address the following issues:
- Thepaper is primarily based on the “Methodology of the 25th of April 184 2016 for the Analysis of a Territory’s Tourist Potential”. This context and content is not explained and since this is probably a local policy text it would be necessary to have some more information. It is suggested to at least briefly present its main points, especially for the readers that are not familiar with the planning regime in Romania.
- The presentation of the methodology of the survey can be improved, the reader doesn’t understand whether hypotheses have been made and have been met, the key questions should be highlighted in a stronger way.
- Part 5 of the article is far too extended (from page 11 till 24) and should be shortened: some pictures could be deleted, and some information merged so that the reader can detect the key points.
- The most important weakness of the paper is detected in part 6: the final discussion adds new information and is not reflecting on the initial research questions. To some extent it is also misleading as it doesn’t’ refer to the main points of the paper. My suggestion would to rewrite this part and focus on the take away messages from applying this method, include policy suggestions and propose further research if necessary.
Hope the above is clear and helpful, wishing you all the best.
Answer:
- A synthetized explanation of the national methodology was added at the Materials and Methods chapter.
- Hypotheses have been added at the end of the Introduction chapter
- The Tourism Models Proposals was changed into an individual chapter so that it extends from pages 18-26. The information is relevant for motivating the model profile.
- The irrelevant information was removed from the conclusions and the latter were re-organized with relevant information. We consider that the three tourism models can stand as a recommendation for the public authorities when upgrading the further version of the commune’s strategy. For Cetariu commune a project was created for the capitalization of the ecotourism and rural potential resulting in plans, drafts and orientations in terms of the large and small tourist infrastructure at a local level which appear in reference no. 51 form the bibliography list. Further research proposals relate to the tourism impact on the commune’s host society, the impact of tourism on rural OMA demographic resources relating marginal villages and their revitalisation through tourism.

Reviewer 3 Report
The research is of little interest, for two reasons:
a) Addresses a restricted territorial area with very specific characteristics.
b) It applies a methodology restricted in its use to Romania.
These conditions make the work poorly transferable outside the field of study if it is not fully adapted. On the other hand, it is excessively descriptive and long (all parts, including summary and keywords...), many of the particularities that are pointed out can be suppressed or taken to schemes, maps and summary tables.
Regarding the sources, the review of the literature must start from the classic studies of tourism potential (present since the 1960s), since that is the central theme of the research and then apply it to metropolitan areas. On the other hand, the literature review includes processes that can hardly be assumed in post-communist countries, as they differ in many cases, such as the Canadian, Spanish, Portuguese cases..., the comparisons would be valid when establishing tourist potentialities, but not models of metropolitan tourism, for example. Statements should always be supported by bibliographic citations (for example, where does the information on lines 61-66 come from)
There are terminological inconsistencies, throughout the article leisure, recreation and tourism are confused (even in the title), which are not the same thing. On the other hand, there is talk of issues that are not correct if the particularities of Central and Eastern Europe are not taken into account: there is no greater tourist attraction in nearby areas, in which repulsion occurs, except in specific contexts. In addition, it is necessary to consider in depth the particularities of border and multicultural areas, which explain tourism processes in this area (https://doi.org/10.3390/su13105385).
There is no clear objective, because the tourist potential of the communes cannot be a research objective in itself (it is the technical report that promotes the research), but rather the causes that make it so. In this sense, the existing correlations and causality must be contrasted and contextualized (territorial planning and tourism planning are missing as specific contexts).
The structure is very poor. It rightly addresses metropolitan processes, which must be continued (in the same Introduction) by tourism in metropolitan areas (section 3). The scope of study (there is no situation map, nor does the situation appear on the OMA maps) must be brief and concise (so that it is understood), and be within section 2 (Methods and data). The research must be conceived as a case analysis (read about the specific methodology). The applied methodology must be fully described (and contextualized with a literature review), as well as the form of data collection: when? how?... (omitted). The results of previous investigations (3.2) are useful for discussion, but should not be included in full. In the text, the headings Results and Discussion do not include the appropriate content, it must be taken into account that:
a) Results responds to What do we find?
b) Discussion responds to What does what we have found mean? How does it contrast or have similarities with what others found before us?
The authors' conclusions are a summary, and should conclude on what was found, what was not.
To validate the work, they must redo it completely, set some research objectives (research question, hypothesis) and develop a complete methodology. The results must be synthetic (aimed at a scientific reader) and the discussion rich (it cannot be done if the appropriate topics are not identified).
Author Response
The research is of little interest, for two reasons:
- Addresses a restricted territorial area with very specific characteristics.
Although it addresses a specific area, we deem it is important as a reference for the OMA policy makers and researchers who are interested in the topic. The gap in literature is bridged by this study, as, due to the fact that the OMA is a relatively new structure research studies about it are missing from literature.
- It applies a methodology restricted in its use to Romania.
These conditions make the work poorly transferable outside the field of study if it is not fully adapted. On the other hand, it is excessively descriptive and long (all parts, including summary and keywords...), many of the particularities that are pointed out can be suppressed or taken to schemes, maps and summary tables.
The methodology is used in Romania but is transferable to any other territory for study and research. A recent research article, co-authored by Tatar C., using partially the same methodology has been applied to the Lake Tana Territory of Ethiopia for a paper in course of publication at the ICON Best Conference Proceedings of Skopje of Endalew et al.
The irrelevant descriptive parts of the paper have been removed (mainly at the conclusion). We deem as pertinent both the statistical interpretation and the tourism models proposals as a basis for future tourism research studies, policy makers or entrepreneurs.
The paper already has many themed maps, tables and figures, we deem a generous discussion section counterbalance as appropriate.
Regarding the sources, the review of the literature must start from the classic studies of tourism potential (present since the 1960s), since that is the central theme of the research and then apply it to metropolitan areas. On the other hand, the literature review includes processes that can hardly be assumed in post-communist countries, as they differ in many cases, such as the Canadian, Spanish, Portuguese cases..., the comparisons would be valid when establishing tourist potentialities, but not models of metropolitan tourism, for example. Statements should always be supported by bibliographic citations (for example, where does the information on lines 61-66 come from)
The in-text citation 5 is given for the distance-decay concept which relates to the lines 61-65, i.e. McKercher, B The Impact of Distance on Tourism: A Tourism Geography Law. In Tourism Spaces, Environments, Locations, and Movements; Lew, A. A. (Ed); Rouletge: London, New York, 2022; pp. 137–141.
There are terminological inconsistencies, throughout the article leisure, recreation and tourism are confused (even in the title), which are not the same thing. On the other hand, there is talk of issues that are not correct if the particularities of Central and Eastern Europe are not taken into account: there is no greater tourist attraction in nearby areas, in which repulsion occurs, except in specific contexts. In addition, it is necessary to consider in depth the particularities of border and multicultural areas, which explain tourism processes in this area (https://doi.org/10.3390/su13105385).
The reference of Wendt et al. (2021) has been cited according to the suggestion at lines 839-840.
There is no clear objective, because the tourist potential of the communes cannot be a research objective in itself (it is the technical report that promotes the research), but rather the causes that make it so. In this sense, the existing correlations and causality must be contrasted and contextualized (territorial planning and tourism planning are missing as specific contexts).
The objectives and work hypotheses have been reformulated and appear at the end of introduction. The data results allowed the elaboration of the three tourism models proposals based on specific item scores from the applied methodology. These are the theoretical implications emerged from the statistical data which can be beneficial for both the private, public sector and researchers. The study also indicates a tourist overconsumption and production over a very small territory (Sanmartin commune) and thus through the models it proposes to divert specific segments of tourists to other communes for a sustainable tourism.
The structure is very poor. It rightly addresses metropolitan processes, which must be continued (in the same Introduction) by tourism in metropolitan areas (section 3). The scope of study (there is no situation map, nor does the situation appear on the OMA maps) must be brief and concise (so that it is understood), and be within section 2 (Methods and data). The research must be conceived as a case analysis (read about the specific methodology). The applied methodology must be fully described (and contextualized with a literature review), as well as the form of data collection: when? how?... (omitted). The results of previous investigations (3.2) are useful for discussion, but should not be included in full. In the text, the headings Results and Discussion do not include the appropriate content, it must be taken into account that:
- Results responds to What do we find?
The results and discussions have been reorganized into a single heading. The results are clearly stated at this section as well as discussions include correlations with other studies, negative consequences which can occur if flows are not properly managed and directed towards other communes to reduce pressure on the small territory of the Sanmartin commune. Discussions are in compliance with the hypotheses formulation.
- b) Discussion responds to What does what we have found mean? How does it contrast or have similarities with what others found before us?
That is why it was stated that it is a pioneering study as holistic tourism potential in the rural OMA was not previously approached by other authors (except for those illustrated at section 3.2 but for individual categories) thus making it difficult to compare with what others found about this territory’s tourism potential and the study is meant to bridge this gap in literature.
The authors' conclusions are a summary, and should conclude on what was found, what was not.
Conclusions have been shortened and are more specific.
To validate the work, they must redo it completely, set some research objectives (research question, hypothesis) and develop a complete methodology. The results must be synthetic (aimed at a scientific reader) and the discussion rich (it cannot be done if the appropriate topics are not identified).
Hypotheses and research aim were reformulated and appear at the end of introduction.

Reviewer 4 Report
First of all congratulations to the authors, the work is most original and provides a lot of knowledge. In my opinion, the first thing to do is reduce the size of the keywords, put the authorship in the figures and graphics, which does not appear in any of them. The background of the research will have to be defined as a theoretical framework for the work. A single hypothesis is not proposed to verify the problem that arises, in material and methods the program used to create the figures and to process the data should be discussed. The results do not explain anything, it jumps directly to the discussion. The conclusions are too extensive, perhaps it would be convenient to specify and reduce. Regarding the bibliography, it would be good to attach more jcr level bibliography.
Author Response
First of all congratulations to the authors, the work is most original and provides a lot of knowledge. In my opinion, the first thing to do is reduce the size of the keywords, put the authorship in the figures and graphics, which does not appear in any of them. The background of the research will have to be defined as a theoretical framework for the work. A single hypothesis is not proposed to verify the problem that arises, in material and methods the program used to create the figures and to process the data should be discussed. The results do not explain anything, it jumps directly to the discussion. The conclusions are too extensive, perhaps it would be convenient to specify and reduce. Regarding the bibliography, it would be good to attach more jcr level bibliography.
Answer:
The size of the keywords has been reduced.
A copyright sign has been put on the maps, below each map the indication that the themed maps are the autors’ own elaboration is given through the statement Own elaboration. Putting all authors names on the map would load and meanwhile change the graphics of the map.
The heading Background research was changed into Theoretical Framework.
The hypotheses have been improved and appear at end of the introduction chapter along with the aim.
The used program for the maps elaboration appears at the Metods and Materials chapter.
Two chapters were merged into one as they are interconnected, i.e. Current Research Results and Discussions.
Conclusions have been reorganized and further specificity made relevant to the study.
More jcr references have been added such as:
Turner, L.; Ash, J. ‘The’ golden hordes: international tourism and the pleasure periphery. 1st Ed.; Constable and Robinson Limited, UK, 1975.
Wanhill, S.; Buhalis, D. Introduction: Challenges for tourism in peripheral areas. International Journal of Tourism Research 1999, 1, 295-297.
Ibănescu, B. Les conditions de la mise en tourisme d’un espace rural périphérique de l’Union Européenne. La province de Moldavie en Roumanie. Bordeaux, Université Michel de Montagne Bordeaux III, PhD. Thesis, France, 2012.
Popescu, A.-C. Tourism Development in a Rural Periphery. Case Study: the Sub-Carpathians of Oltenia. Journal of Settlements and Spatial Planning 2014, 3, 53-64.
Lunt, N.; Smith, R.D.; Exworthy, M.; Green, S.T.; Horsfall, D.G.; Mannion, R. Medical Tourism: Treatments, Markets and Health System Implications: A scoping review. OECD, 2011; Gligorijevič, Z.;
Novovič, M. Health and Recreation Tourism in The Development of Mountain Spas and Resorts, Economic Themes 2014, 52(4), 498-512;
Esfandiari, H.; Choobchian, S. Designing a Wellness-Based Tourism Model for Sustainable Rural Development. Research Square 2020, 1-28,
Romão, J.; Machino, K.; Nijkamp, P. Integrative diversification of wellness tourism services in rural areas–an operational framework model applied to east Hokkaido (Japan). Asia Pacific Journal of Tourism Research 2018, 23(7), 734-746;
Winter, P.L.; Selin, S.; Cerveny, L.; Bricker, K. Outdoor Recreation, Nature-Based Tourism, and Sustainability. Sustainability 2020, 12, 81;
Zhao, L.; Wang, X. Rural Housing Vacancy in Metropolitan Suburbs and Its Influencing Factors: A Case Study of Nanjing, China. Sustainability 2021, 13, 3783.

Reviewer 5 Report
It is a good article from the methodological point of view that allows to obtain an evaluation of the tourist potential considering the category of natural attractions, the category of attractions created by man, the category of tourist facilities, the category of technical facilities. However, I consider it necessary to suggest improvements above all to highlight the theoretical implications that the manuscript should have.
I suggest giving more emphasis to the gap in the literature for the manuscript that deals with the evaluation of the tourism potential of a territory. The discussion section focuses more on discussing the results of the evaluation of the tourism potential in the communes of the Oradea Metropolitan Area (OMA) that is important, but the contribution of the manuscript that leads to the theoretical implications is little discussed, for this reason the conclusions little emphasis is placed on the theoretical contribution.
Although the objectives are understood, I consider that it helps to clarify the question of the investigation.
The methodology used and the results obtained with this methodology are adequate and are very well explained in the manuscript, on the tourist potential of the metropolitan communes, but the article must show that it contributes to the literature, so the gap in the literature must be be clear, the discussion should lead to the determination of the theoretical implications and conclusions should emphasize the theoretical contribution of the manuscript to the literature.
On the other hand, I suggest analyzing if it is convenient to place images and map in the discussion section, perhaps it is better to place them in another section.
There are paragraphs that only have 2 lines, perhaps it is better to have at least 4 lines.
The manuscript has as a section 2 materials and methods before Research Background, examine if it is not appropriate to place it after.
Author Response
It is a good article from the methodological point of view that allows to obtain an evaluation of the tourist potential considering the category of natural attractions, the category of attractions created by man, the category of tourist facilities, the category of technical facilities. However, I consider it necessary to suggest improvements above all to highlight the theoretical implications that the manuscript should have.
I suggest giving more emphasis to the gap in the literature for the manuscript that deals with the evaluation of the tourism potential of a territory. The discussion section focuses more on discussing the results of the evaluation of the tourism potential in the communes of the Oradea Metropolitan Area (OMA) that is important, but the contribution of the manuscript that leads to the theoretical implications is little discussed, for this reason the conclusions little emphasis is placed on the theoretical contribution.
Although the objectives are understood, I consider that it helps to clarify the question of the investigation.
The methodology used and the results obtained with this methodology are adequate and are very well explained in the manuscript, on the tourist potential of the metropolitan communes, but the article must show that it contributes to the literature, so the gap in the literature must be be clear, the discussion should lead to the determination of the theoretical implications and conclusions should emphasize the theoretical contribution of the manuscript to the literature.
On the other hand, I suggest analyzing if it is convenient to place images and map in the discussion section, perhaps it is better to place them in another section.
There are paragraphs that only have 2 lines, perhaps it is better to have at least 4 lines.
The manuscript has as a section 2 materials and methods before Research Background, examine if it is not appropriate to place it after.
Answer
For the theoretical implications the three tourism models proposals rely on the emerged results of the assessment study and allowed to highlight which communes can target certain segments of the population based on the identified resources so that tourist flows will spread evenly in the metroplitan area and not exert tourist pressure on a single commune as in the case of Sanmartin. Items of the apllied methodology were also considered for the models elaboration. These models can stand as recommendations for the public authorities when upgarding with newer versions of the the commune’s integrated strategies as well as the study results for further tourism projects. Gaps in the literature referring to the OMA communes tourism constitute a problem that is why this study can be used by local authorities and researchers for further tourism policy making. The fact that the metopolitan areas emerged quite recently in Romania, afther the year 2000, so lacking a tradition from this sense, as before Romanian cities were „closed”, following a strict standardized planning which does not allow its territorial or demographic extension, the gap of references and literature to the metropolitan Romanian areas is accountable.
The section of Materials and Methods has been placed after the Research Background.

Reviewer 6 Report
The title of the article is very interesting and promising. However, the article is a bit chaotic and needs to be organized.
1. The aim of the study is defined differently in different parts of the article (lines: 24-25, 101-104, 142-145 and 170.
The aim of the study should be in the abstract and at the end of the introduction.
2 There is a lot of information in the introduction about urbanization, which is not directly related to the topic of the article and the aim of the study.
3. The information contained in the introduction about the Oradea Metropolitan Area and figure 1 should be in the second chapter: Materials and Methods as a characterization of the study area.
4. The tourism potential that is currently described in the methodology should rather be in the Introduction or Results
5. The methodology should list all the work that was used for the study and which methods
6 Chapter: Current Research Results presents only Table 1. Some description should be included is the data in the table
The conclusions and results are well described.
Author Response
The title of the article is very interesting and promising. However, the article is a bit chaotic and needs to be organized.
The aim of the study is defined differently in different parts of the article (lines: 24-25, 101-104, 142-145 and 170.
The aim of the study should be in the abstract and at the end of the introduction.
2 There is a lot of information in the introduction about urbanization, which is not directly related to the topic of the article and the aim of the study.
- The information contained in the introduction about the Oradea Metropolitan Area and figure 1 should be in the second chapter: Materials and Methods as a characterization of the study area.
- The tourism potential that is currently described in the methodology should rather be in the Introduction or Results
- The methodology should list all the work that was used for the study and which methods
6 Chapter: Current Research Results presents only Table 1. Some description should be included is the data in the table
The conclusions and results are well described.
Answers
- The aim has been reformulated and appears unitary at the abstract section and at the end of introduction.
- The exceeding urbanization information has been removed.
- We consider that the study area and map should remain in the introduction section as it was contextualized within this section in a general to particular approach.
- The tourism potential literature has been placed at the introduction section
- The methodology was reorganised and new information methodological information was added
- Current research results chapter eas merged to the Discussions chapter as they are interconnected and thus onle single braod chapter resulted entitled Current research results and discussions

Reviewer 7 Report
The study is interesting and provide insights into the tourism aspects in a well-known touristic area in Romania, with practical implications for the local tourism industry. The presentation is clear and well formulated. The study is basically a summary of previous studies of at least some of the authors, while the novelty resides mainly in the discussions of tourism models. However, some revisions would be useful for improving the clarity and accessibility of the material, for which suggestions are presented in the following.
Line 181: please explain WTO acronyms and, if possible, provide a link to the definition cited.
Lines 212-251: Please provide the list of previous studies used and referred to in this paragraph. Apparently, the list (in fact a description of the results of those studies) is presented in section 3.2. If so, please rephrase the current (212-215) paragraph.
Lines 272-279: A reformulation of this paragraph may be useful, as it is not clear e.g. who recommends and in what context/conditions the Chines village model development etc.
Line 527: A more detailed description of the statistical analysis listed here should be provided such that to support the results on the tourism models identified/described.
Author Response
The study is interesting and provide insights into the tourism aspects in a well-known touristic area in Romania, with practical implications for the local tourism industry. The presentation is clear and well formulated. The study is basically a summary of previous studies of at least some of the authors, while the novelty resides mainly in the discussions of tourism models. However, some revisions would be useful for improving the clarity and accessibility of the material, for which suggestions are presented in the following.
Line 181: please explain WTO acronyms and, if possible, provide a link to the definition cited.
The citation was provided and acronyms explained.
Lines 212-251: Please provide the list of previous studies used and referred to in this paragraph. Apparently, the list (in fact a description of the results of those studies) is presented in section 3.2. If so, please rephrase the current (212-215) paragraph.
The paragraph was rephrased and the section where these references appear was mentioned.
Lines 272-279: A reformulation of this paragraph may be useful, as it is not clear e.g. who recommends and in what context/conditions the Chines village model development etc.
The rephrase was done accordingly: The effect of all the influences and actors involved in tourism emerges from the proposed models of interaction and spatial distribution such as the Chinese village development model for tourism such as a market-based, traffic-oriented and scenic-radiant source.
Line 527: A more detailed description of the statistical analysis listed here should be provided such that to support the results on the tourism models identified/described.
The applied statistical items for the tourism models elaboration were explained at the beginning of the chapter 5 (Tourism Models Proposals).

Round 2
Reviewer 2 Report
Thank you for the feedback to my comments
Author Response
Dear Prof.,
Thank you for the feedback to my comments.
Answer. We thank you for the pertinent advice that led us to the completion of this study.

Reviewer 3 Report
The authors do not understand what a revision is, not facing most of the changes that are proposed to it. It is not a matter of discussing what the reviewer tells them, but of doing it and, if it is not possible, explaining why it cannot be done. In this way, if it is argued that the work is of little interest because it is a specific methodology for a specific place, the authors cannot say that since it is a specific methodology and a specific case it cannot be discussed. The authors are neither the first to have studied metropolitan tourism nor the tourist potential in metropolitan areas; they may be the first to study OMA, but for a top-level international journal it is only interesting as a case study, and if the work cannot be transferred elsewhere, due to methodological development and discussion, it should be sent to a national and/or regional magazine, in which you are sure to be interested.
What is asked of the authors is that they read about the topics that are addressed, and from them be able to discuss and conclude, to contribute to the scientific field.
The hypothesis is not true as a starting point and it is shown in figure 7. Sanmartin concentrates more resources, and has been more favored, so it cannot be said that the potential of all is the same, there are more variables, which must be addressed; but if he doesn't check the literature they can't establish it. They come up with a problem, but they don't deal with it properly. The objective cannot be to highlight the tourist potential, because it is part of the technique (assessment for planning) and not science (why assess in that way to avoid problems... and that it be transferable to other places). Read about tourism potential and from there establish research questions, hypotheses and objectives.
Its methodology responds to a case analysis, contextualize. The methodology is not that of the previous technical work, it is that of this work that is presented here, they must not only tell how the tourist potential is inventoried and established, but also the development of the scientific work (adjust to this: the technical report, the document scientific).
It continues to have serious structural deficiencies, since the authors have not made all the changes that this reviewer proposed (the scope of study is still in the introduction, which should be devoted exclusively to the background).
Still no location map and maps still don't show location. The map can be from OMA or from the peri-urban area of any other central European city and, in the scientific community, I express my doubts that everyone will identify where Oradea is and its position in Romania. From a purely cartographic perspective, the correctness of the maps requires situation.
The authors continue to insist on comparisons with the environment that have nothing to do with it, do not make extensions of what happens in other places. Review the literature in depth and in the discussion see what can or cannot be found, transferred from the readings made, that is correlating.
The conclusions are still descriptive, and a summary, when they must answer the research questions and the hypotheses raised.
In the first revision they are told that it is a long document, they are proposed to eliminate the remaining elements; your review is 4 pages longer than the previous one. It is about inserting the work within a scientific context (reading about tourism, tourism potential...) and eliminating everything that is inconsequential (particularities that are only understandable in a local or regional context).
Author Response
Dear Prof.,
We have tried to follow your advice as closely as possible, taking into account and respecting the opinions of the other anonymous reviewers appointed by SUSTAINABILITY, so as not to conflict with their thematic expectations.
We are honored to have had you as a reviewer and we were able to share your knowledge. We were the ones who won including through the deep manifestation of your critical and analytical spirit.
Below, we have explained how the interventions for qualitative enhancement were made in the text of the article, trying to follow your requests as closely as possible. We thank you for everything.
The authors do not understand what a revision is, not facing most of the changes that are proposed to it. It is not a matter of discussing what the reviewer tells them, but of doing it and, if it is not possible, explaining why it cannot be done. In this way, if it is argued that the work is of little interest because it is a specific methodology for a specific place, the authors cannot say that since it is a specific methodology and a specific case it cannot be discussed. The authors are neither the first to have studied metropolitan tourism nor the tourist potential in metropolitan areas; they may be the first to study OMA, but for a top-level international journal it is only interesting as a case study, and if the work cannot be transferred elsewhere, due to methodological development and discussion, it should be sent to a national and/or regional magazine, in which you are sure to be interested.
The applied methodology is a law published in the Romanian Official Monitory 444 on the 14th of June 2014 and we applied this methodology at the case study of OMA, which appears at the reference List at [55]. It is a transferrable methodology, all Romania’s territory has been assessed through it to find out its tourist potential. But it can easily be applied to any other territory beyond Romania’s borders. It can be applied to any high or small-scale territory. The paper’s aim is to highlight its resulted tourist models which can appleal to particular market segments and the OMA communes tourist potential assessment is a tool through which we were able to propose these models.
What is asked of the authors is that they read about the topics that are addressed, and from them be able to discuss and conclude, to contribute to the scientific field.
We consider that we contribute to the scientific field by the elaboration of the models which can inspire other international tourist metropolitan studies as well as the elaboration of local tourism policy makers.
The hypothesis is not true as a starting point and it is shown in figure 7. Sanmartin concentrates more resources, and has been more favored, so it cannot be said that the potential of all is the same, there are more variables, which must be addressed; but if he doesn't check the literature they can't establish it. They come up with a problem, but they don't deal with it properly. The objective cannot be to highlight the tourist potential, because it is part of the technique (assessment for planning) and not science (why assess in that way to avoid problems... and that it be transferable to other places). Read about tourism potential and from there establish research questions, hypotheses and objectives.
The paper’s aim is to highlight its resulted tourist models which can appleal to particular market segments and the OMA communes tourist potential assessment is a tool through which we were able to propose these models. Figure 7 shows that there are tourist attractions within all the OMA communes but unevenly territoreially distributed, which also highlights hypothesis no 2, i.e. the second hypothesis starts form the premise that all communes provide a tourist supply for a segmented market, but which is unevenly dispersed in the OMA communes.
Its methodology responds to a case analysis, contextualize. The methodology is not that of the previous technical work, it is that of this work that is presented here, they must not only tell how the tourist potential is inventoried and established, but also the development of the scientific work (adjust to this: the technical report, the document scientific).
The methodology results have been taken over from previously individually published papers on the assessment of natural resources from reference of Tătar et al [2018], man made resources from reference no Tatar et al [2021], tourist and technical facilities Stasac et al. [2020]. All these four categories’ results were synthetized in our paper so as to get a final holistic score from all these four perspectives for each OMA commune.
The methodology is indeed better suited for case studies, as that of OMA, but it can be applied to any other high or small-scale territory too. In the published book of Corina Tatar (2009) Posibilitati de Valorificare a resurselor turistice din bazinul hidrografic al Crisului Negru/ Tourist Resources Capitalization Opportunities of the Catchment Area of Crisul Negru River used the same methodology for an extended territory. The development to the scientific field resides in the models elaboration by using the variables of the methodology explained at the fifth chapter initial part, tourism proposal models.
It continues to have serious structural deficiencies, since the authors have not made all the changes that this reviewer proposed (the scope of study is still in the introduction, which should be devoted exclusively to the background).
The aim and hypotheses have been reformulated and moved to the third section Materials and Methods.
Still no location map and maps still don't show location. The map can be from OMA or from the peri-urban area of any other central European city and, in the scientific community, I express my doubts that everyone will identify where Oradea is and its position in Romania. From a purely cartographic perspective, the correctness of the maps requires situation.
Figure no 1. has been reconfigured and now shows the rural OMA location at the level of Romania, it appears at the introductory section.
The authors continue to insist on comparisons with the environment that have nothing to do with it, do not make extensions of what happens in other places. Review the literature in depth and in the discussion see what can or cannot be found, transferred from the readings made, that is correlating.
The environmental related parts have been removed from the text, despite the fact that we deem that flows overconcentration as in the case of Sanmartin can lead and has led to environmental damage. The environmental related extinction of the vegetal, fish and nail species were a direct consequence of such human pressure on the small commune of Sanmartin. That is why tourist flows need to be dispersed to surrounding communes and tourism models were proposed for this reason.
The conclusions are still descriptive, and a summary, when they must answer the research questions and the hypotheses raised.
At the conclusion section the hypotheses questions were answered in the second, third and fourth papargraph of the concusion section (i.e. 6) and the models relevance also referred to.
In the first revision they are told that it is a long document, they are proposed to eliminate the remaining elements; your review is 4 pages longer than the previous one. It is about inserting the work within a scientific context (reading about tourism, tourism potential...) and eliminating everything that is inconsequential (particularities that are only understandable in a local or regional context).
We have eliminated the part 2.2. Previous tourism research results of the constituent communes of the Oradea Metropolitan Area and reformulated it into a small paragraph. After all changes, the final manuscript has now 29 pages.
Kind regards,
Corina-Florina TATAR, Iulian DINCA, Ribana LINC, Marius STUPARIU, Liviu BUCUR, Marcu STASAC, Stelian NISTOR

Reviewer 5 Report
The manuscript has improved, it has a better review of the literature, it shows the gap more clearly. The methodology and data processing are consistent. Likewise the discussion
The conclusions express the theoretical implications of the three tourism models proposals rely on the emerged results of the assessment study and allowed to highlight which commons can target certain segments of the population based on the identified resources so that tourist flows will spread evenly in the metropolitan area.
Author Response
The manuscript has improved, it has a better review of the literature, it shows the gap more clearly. The methodology and data processing are consistent. Likewise the discussion
The conclusions express the theoretical implications of the three tourism models proposals rely on the emerged results of the assessment study and allowed to highlight which commons can target certain segments of the population based on the identified resources so that tourist flows will spread evenly in the metropolitan area.
Dear Professor,
We are honored to have had you as an anonymous reviewer for our study. We thank you for the very useful thematic guidelines in the completion of the article.

Reviewer 6 Report
Thank you for your contribution to the improvement of the article. I accept the article for publication in the current version.
Author Response
Thank you for your contribution to the improvement of the article. I accept the article for publication in the current version.
Dear Prof.,
We thank you for the useful advice that guided us in improving the quality of the article.

Round 3
Reviewer 3 Report
After three revisions the authors are still "interpreting" what they are told, and arguing again. The article lacks interest for an international journal:
a) There is no analysis of the scientific literature that contextualizes the studies of tourist potentiality, which is limited to paragraph 70-82, when it comes to the central issue. If what they want is to talk about a methodology, when they state that they want to make a methodology extensible, it is necessary to analyze the methodologies proposed in the literature, they do not.
b) They do not know how to distinguish between the methodology of the previous study (tourist potential) and the methodology of the article. It is a case analysis, and should be contextualized as such. This is where the field of study should appear.
c) Figure 1 is of poor quality, the formal aspects must be taken care of when you want to publish in an impact journal.
d) There are parts out of focus (Theoretical framework), and what they have introduced again does not improve.
e) Attending revision 2 of this reviewer does not mean forgetting revision 1, which they did not attend to in what was requested.
f) There is still no correlation between the parts of the work. The discussions are limited, because what the authors consider important are the results, and this is not the case. Similarly, the conclusions are limited, leaving no clear limitations, replicability, or contribution...
